# IDEAL: Data Equilibrium Adaptation for Multi-Capability Language Model Alignment

**Chenlin Ming[1], Chendi Qu[1], Xiaoming Duan[1], Qizhi Pei[4], Zhuoshi Pan[3], Yu Li[2]**
**Conghui He[2], Lijun Wu[2]***

[1]Shanghai Jiao Tong University      [2]Shanghai Artificial Intelligence Laboratory
[3]Tsinghua University      [4]Renmin University of China
{mcl2019011457,xduan}@sjtu.edu.cn      {wulijun}@pjlab.org.cn

## ABSTRACT

Large Language Models (LLMs) have achieved impressive performance through Supervised Fine-tuning (SFT) on diverse instructional datasets. When training on multiple capabilities simultaneously, the mixture training dataset, governed by volumes of data from different domains, is a critical factor that directly impacts the final model's performance. Unlike many studies that focus on enhancing the quality of training datasets through data selection methods, few works explore the intricate relationship between the compositional quantity of mixture training datasets and the emergent capabilities of LLMs. Given the availability of a high-quality multi-domain training dataset, understanding the impact of data from each domain on the model's overall capabilities is crucial for preparing SFT data and training a well-balanced model that performs effectively across diverse domains. In this work, we introduce **IDEAL**, an innovative data equilibrium adaptation framework designed to effectively optimize volumes of data from different domains within mixture SFT datasets, thereby enhancing the model's alignment and performance across multiple capabilities. IDEAL employs a gradient-based approach to iteratively refine the training data distribution, dynamically adjusting the volumes of domain-specific data based on their impact on downstream task performance. By leveraging this adaptive mechanism, IDEAL ensures a balanced dataset composition, enabling the model to achieve robust generalization and consistent proficiency across diverse tasks. Experiments across different capabilities demonstrate that IDEAL outperforms conventional uniform data allocation strategies, achieving a comprehensive improvement of approximately 7% in multi-task evaluation scores. Our code is available at https://github.com/ming-bot/IDEAL.

## 1 INTRODUCTION

Recent advancements in LLMs have demonstrated their remarkable ability to master diverse capabilities Dong et al. (2024); Zhang et al. (2024b); Hu et al. (2023); Mecklenburg et al. (2024); Li et al. (2025) through Supervised-Fine-tuning (SFT) on instruction-aligned datasets Liu et al. (2023); Lu et al. (2023); Agarwal et al. (2024); Wang et al. (2023). By training on heterogeneous tasks such as mathematical reasoning Luong et al. (2024); Guo et al. (2025); Pei et al. (2025); Pan et al. (2025), code generation Dehaerne et al. (2022); Si et al. (2024), and creative writing Wang et al. (2024); Gómez-Rodríguez & Williams (2023); Franceschelli & Musolesi (2024), models like GPT-4 OpenAI et al. (2024), Claude Anthropic (2023), and LLaMA-3 Grattafiori et al. (2024) achieve promising performance across various domains. However, empirical studies reveal that naively merging datasets for multi-objective fine-tuning often degrades performance compared to single-task specialization Tunstall et al.; Shen (2024); Dong et al. (2024). To mitigate the aforementioned issue, a common approach is to adjust the training data distribution Xie et al. (2024); Ye et al. (2024), thereby regulating the volume of data from each domain within the mixed dataset. However, critical challenges persist: the optimal mixture proportions of these domains are poorly understood and how to adjust the optimal mixture proportions is not clear. While heuristic solutions such as manual data

---

*Corresponding author

reweighting or rule-based curriculum learning Bengio et al. (2009) exist, they suffer from scalability limitations and suboptimal task balance. Prior attempts to automate data allocation, including pretraining-centric methods Xie et al. (2024); Ye et al. (2024), fail to address the unique dynamics of SFT, where data-task alignment directly governs cross-domain interference. Consequently, a principled framework for resolving data conflicts in multi-capability SFT remains an open problem.

To address this challenge, we propose **IDEAL**, a novel framework that dynamically aligns SFT data mixtures with model capabilities. IDEAL employs the second-order information to optimize the data mixture ratios iteratively. Unlike previous works using the influence function for data sample selection Xia et al. (2024); Zhang et al. (2024a), we employ a domain-specific hyperparameter to regulate the volume of training data for each domain. To scale our method to large model parameter sizes and ensure computational efficiency, we optimize the approach for efficiently computing the Hessian matrix and incorporate upsampling and downsampling strategies, significantly speeding up the process of adjusting training data distributions. Through iterative optimization of the data volume for each domain, the model trained on the optimized dataset exhibits a significant

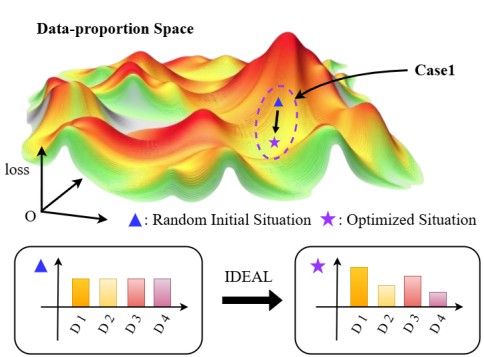

Figure 1: IDEAL adjusts data mixture proportions to optimize model performance, leading to a decrease in loss.

reduction in loss, as shown in Fig. 1. It's important to clarify that a balanced dataset does not simply mean having equal amounts of data from each domain, but rather finding an optimal distribution where all capabilities can develop properly. This model-aware mechanism adapts to the LLM's evolving training dynamics, ensuring equilibrium between data efficiency and task balancing. Crucially, our framework operates without costly hyperparameter sweeps, enabling scalable multi-capability SFT with theoretical guarantees.

Extensive experiments validate IDEAL's effectiveness across diverse capability combinations, including reasoning, mathematics, code generation, and instruction-following. On multiple benchmarks, IDEAL outperforms uniform data blending by 7% on average and even surpasses single-task SFT baselines after 2 iterations, demonstrating its ability to resolve data conflicts while amplifying cross-task synergies. Further analysis delves into a sensitivity study on the IDEAL algorithm's performance with the selection of hyperparameters and its robustness on extended benchmarks and domains, all of which help in comprehensively evaluating the proposed method. These results establish data equilibrium adaptation as a critical lever for training generalist LLMs, advancing beyond the trial-and-error paradigm of conventional SFT. IDEAL offers a practical, mathematically grounded method for LLM data optimization, bridging innovation with real-world applicability.

## 2 RELATED WORK

**Data Mixing.** Techniques like data mixing, alternatively referred as data re-weighting, center on refining the distribution of characteristics within a pre-prepared dataset, which aims to enhance models' multi-capability performance. Conventional methods typically depend on approaches based on the ratio of tokens Touvron et al. (2023); Shen et al. (2024); Liu et al. (2025). Some methods also focus on data selection to change the distribution of training data by selecting high-quality training data Parmar et al. (2024); Chung et al. (2023); Engstrom et al. (2024); Xia et al. (2024); Mekala et al. (2024); Kang et al. (2024) Xu et al. (2023b). In the LLM era, data mixing has evolved into learning-based approaches by using optimization algorithms or some smaller well-trained models Chen et al. (2024); Shao et al. (2024); Yu et al. (2024). Wu Wu et al. (2024) employs a reinforcement learning approach to train a scorer network for data sampling probability allocation, constituting a bi-level data selection framework. Dong Dong et al. (2024) investigates the DMT training procedure, manually designing mixing ratios to demonstrate multi-domain training challenges. DoReMi Xie et al. (2024) employs group distributionally robust optimization to fine-tune a small proxy model. After determining the optimal proportions, DoReMi transfers them to a larger LLM, significantly reduc-

ing training costs. DOGE Fan et al. (2023) builds on this approach by re-weighting the training dataset for specific target domains, also utilizing a trained proxy model to determine the final domain weights. Data Mixing Law Ye et al. (2024) introduces a mixing proportion law based on extensive experimentation. The mixing proportion law reveals the relationship between data proportions and the corresponding validation loss, providing a reference for adjusting the data distribution. However, these methods require costly global weight searches and often disregard the continuity of data distributions. Our work addresses these gaps by gradient-guided iterative refinement, enabling efficient adaptation.

**Data Selection.** Data quality is a decisive factor for model performance Albalak et al. (2024). Data selection techniques are widely used to identify beneficial data from massive datasets for model training, particularly in the pre-training phase Gu et al. (2022). LESS Xia et al. (2024) leverages first-order gradient information to compare data gradients with target gradients, selecting data that aligns more closely with the target distribution. Gu et al. Gu et al. (2024) model the training process as a dynamical system, applying Pontryagin's Maximum Principle (PMP) Pontryagin (2018) to optimize data selection for training. Fan et al. Fan et al. (2025) focuses on domain similarity to address dimensional collapse, introducing the DiSF method to enhance data diversity in training. Liu et al. Liu et al. (2024) proposes SelectIT, which utilizes the internal uncertainty signals of LLMs to select high-quality data, eliminating the need for external resources. However, most data selection methods assume the availability of large-scale, reliable data and have not yet been extended to the SFT phase, where high-quality data is often scarce.

**Influence Function.** The Influence Function Hampel (1974) has undergone decades of development and has attained widespread recognition due to its remarkable interpretability. In learning-based methods, the influence function, as noted in Koh & Liang (2017), serves as a crucial link between model performance and the training dataset. It casts light on the inner workings of the black-box predictor, demystifying the often opaque process of the model behavior. Despite the high cost of computational resources Grosse et al. (2023); Ilyas et al. (2022); Zhang et al. (2024a), many gradient-based approximation methods are used to explore the relationship between the performance of the large language model and the training set Engstrom et al. (2024); Park et al. (2023); Pruthi et al. (2020). Liang Liang et al. (2025) uses an iterative train-filter process and focuses on instance-level data filtering. MATES Yu et al. (2024) utilizes a proxy model to estimate the influence score for selecting the pre-training dataset. However, influence-based methods are overly focused on individual data points and lack the ability to handle mixed datasets effectively. While they are well-suited for filtering data on a per-instance basis, they are not efficient for tasks that require adjusting the overall distribution of a dataset.

# 3 METHODOLOGY

## 3.1 PROBLEM FORMULATION

We first give the necessary notations and problem formulation. Assume a base model $\mathcal{M}_0$ parameterized with $\boldsymbol{\theta} \in \mathbb{R}^M$ exists, and our objective is to leverage SFT to develop an instruction-tuned model based on $\mathcal{M}_0$ that excels across multiple domains. To facilitate the optimization of the whole training dataset $\mathcal{D}_{tr}$, we partition the training data into distinct categories based on their respective domains, such as math, code, and so on, as $\mathcal{D}_1, \ldots, \mathcal{D}_n$. Let $N$ be the total number of training samples $N = |\mathcal{D}_{tr}| = |\mathcal{D}_1| + \cdots + |\mathcal{D}_n|$, where $|\mathcal{D}_i|$ represents the cardinality of the domain-specific training dataset $\mathcal{D}_i$. The learning objective is to optimize model parameters by minimizing the training cost function, with optimal parameters expressed as:

$$\boldsymbol{\theta}^* = \arg\min_{\boldsymbol{\theta}} \mathcal{L}(\mathcal{D}_{tr}, \boldsymbol{\theta}) = \arg\min_{\boldsymbol{\theta}} \frac{1}{N} \sum_{i=1}^{n} \mathcal{L}(\mathcal{D}_i, \boldsymbol{\theta}),$$

where $\boldsymbol{\theta}$ is the parameter of the model, and $\mathcal{L}$ is the parameterized empirical cost function. We employ a small, independent reference dataset $\mathcal{D}_{ref}$ to rigorously validate its performance across diverse domains. This dataset is strictly excluded from the training process to ensure unbiased and fair assessment. The cost of the well-trained model on $\mathcal{D}_{ref}$ serves as a comprehensive performance metric across multiple tasks, and our ultimate objective is to minimize the reference loss: $\mathcal{L}(\mathcal{D}_{ref}, \theta^*)$.

### 3.2 Optimization of Multi-domain Training Dataset

When simply mixing the data from multiple domains together, like $\mathcal{D}_{tr} = \mathcal{D}_1 \cup \cdots \cup \mathcal{D}_n$ for SFT, the instruction-tuned model is easy to show imbalanced capabilities (as shown in Table 1). Although each domain-specific training set has been individually validated for effectiveness, a straightforward mixture of these datasets could result in an inappropriate training data distribution. The unbalanced blending of data volumes across domains may introduce data conflicts, causing the trained model to fail in mastering specific domain tasks. One direct approach is to expand the training data (e.g., adding new training data) for underperforming capabilities while reducing the data volume for other domains. However, determining the precise amount of data to add or remove remains an unclear problem, which often lacks a scientific basis and frequently relies on the empirical judgment of engineers. Therefore, introducing a principled approach for determining the data mixture ratio is a crucial task. Besides, according to Muennighoff's findings Muennighoff et al. (2023), repeating existing data up to four times yields a performance improvement that is largely equivalent to introducing the same amount of new data, which further supports the rationality of modifying the multi-domain data mixture ratio. Building on this insight, we introduce a parameter $\boldsymbol{\beta} \in \mathbb{R}^n$ to control the proportion of repeated data relative to the original domain-specific training set, thereby dynamically adjusting the volume of data from different domains. To achieve this, we redefine the learning function and optimal parameters as:

$$\boldsymbol{\theta}^* = \frac{1}{N + \sum_{i=1}^{n} \beta_i |\mathcal{D}_i|} \arg\min_{\boldsymbol{\theta}} \left( \mathcal{L}(\mathcal{D}_{tr}, \boldsymbol{\theta}) + \sum_{i=1}^{n} \beta_i \mathcal{L}(\mathcal{D}_i, \boldsymbol{\theta}) \right). \tag{1}$$

Note that $\beta_i$ is a real number that can be either positive or negative; when $\beta_i$ is positive, it indicates the need to repeat existing data, whereas when it is negative, it signifies the requirement to reduce the volume of the corresponding domain-specific data. Finding the optimal proportion $\boldsymbol{\beta}$ is equal to solving the following bi-level optimization problem:

$$\min_{\boldsymbol{\beta} \in \mathcal{S}} \mathcal{Q}(\boldsymbol{\beta}) := \mathcal{L}(\mathcal{D}_{ref}, \boldsymbol{\theta}^*)$$

$$s.t. \text{ Eq. (1)}. \tag{2}$$

Our objective is to understand the impact of $\boldsymbol{\beta}$ on the model's performance on $\mathcal{D}_{ref}$ and find the best optimization direction for $\boldsymbol{\beta}$. By applying the chain rule, we determine the impact of a specific $\beta_j \in \{\beta_1, \ldots, \beta_n\}$ on the model's performance $\mathcal{L}(\mathcal{D}_{ref}, \boldsymbol{\theta}^*)$ as:

$$\frac{\partial \mathcal{Q}(\boldsymbol{\beta})}{\partial \beta_j} = \left( \frac{\partial \mathcal{L}(\mathcal{D}_{ref}, \boldsymbol{\theta}^*)}{\partial \boldsymbol{\theta}^*} \right)^\top \frac{\partial \boldsymbol{\theta}^*}{\partial \beta_j}. \tag{3}$$

**Assumption 1** *The empirical risk function $\mathcal{L}$ is twice differentiable with respect to $\boldsymbol{\theta}$. $\mathcal{S}$, defined as the domain of $\boldsymbol{\beta} \in \mathbb{R}^n$, is a compact and convex set.*

This assumption is readily applicable to the general training of LLMs where cross-entropy is used as the loss function. With this assumption in place, we can derive the following lemma.

**Lemma 1** *Suppose Assumption 1 holds. The impact of a specific $\beta_j$ on the optimal model parameters $\boldsymbol{\theta}^*$ trained on the whole training set can be explicitly expressed as Eq. (4).*

$$\frac{\partial \boldsymbol{\theta}^*}{\partial \beta_j} = - \left[ \nabla^2 \mathcal{L}(\mathcal{D}_{tr}, \boldsymbol{\theta}^*) + \sum_{i=1}^{n} \beta_i \nabla^2 \mathcal{L}(\mathcal{D}_i, \boldsymbol{\theta}^*) \right]^{-1} \nabla \mathcal{L}(\mathcal{D}_j, \boldsymbol{\theta}^*). \tag{4}$$

For a comprehensive derivation and detailed proof, please refer to Appendix B.

Looking back into the initial training configuration, we employ the straightforwardly combined dataset to refine the model $\mathcal{M}_0$ into $\mathcal{M}_1$, thereby obtaining the optimized model parameters $\boldsymbol{\theta}^*$. With the initial state $\boldsymbol{\beta} = (0, \ldots, 0)$, we can therefore obtain the influence of $\boldsymbol{\beta}$ on the validation set $\mathcal{D}_{ref}$ as Eq. (5) by substituting Eq. (4) into Eq. (3).

$$\left. \frac{\partial \mathcal{Q}(\boldsymbol{\beta})}{\partial \boldsymbol{\beta}} \right|_{\boldsymbol{\beta}=(0,\ldots,0)} = - \left( \frac{\partial \mathcal{L}(\mathcal{D}_{ref}, \boldsymbol{\theta}^*)}{\partial \boldsymbol{\theta}^*} \right)^\top \nabla^2 \mathcal{L}(\mathcal{D}_{tr}, \boldsymbol{\theta}^*)^{-1} \nabla \mathcal{L}(\mathcal{D}_j, \boldsymbol{\theta}^*). \tag{5}$$

### 3.3 EFFICIENT CALCULATION

Evaluating the inverse of the Gauss-Newton Hessian in Eq. (5) for a large language model presents a formidable challenge. In this section, we employ approximation methods and acceleration techniques to achieve efficient computations. According to K-FAC theory Martens & Grosse (2015); Ueno et al. (2020); Zhang et al. (2024a), we use the Kronecker product to accelerate the inverse Hessian-vector products (iHVP) computation. Considering minimal direct dependencies between the gradients of different MLP layers, we decompose the Hessian matrix $\mathbf{H}$ into a simple block-diagonal approximation according to different layers. Specifically, we utilize the hook function to obtain the gradients of each layer. The gradient with respect to layer $l$ can be expressed as $D\theta^l = \text{vec}(DW^l) = x^l \otimes \delta^l$. Here, $D\theta^l$ represents the gradient of the parameters in layer $l$, which is vectorized as $\text{vec}(DW^l)$. The term $x^l$ denotes the input to layer $l$, and $\delta^l$ represents the error gradient propagated back to this layer. The symbol $\otimes$ indicates the outer product operation, which combines $x^l$ and $\delta^l$ to compute the gradient update for the parameters in layer $l$. Thus, the block matrix of the $l$ layer in the Hessian matrix $\mathbf{H}$ is:

$$
\begin{aligned}
\mathbf{H}^l &= \mathbb{E}(D\theta^l D\theta^{l\top}) = \mathbb{E}(x^l x^{l\top} \otimes \delta^l \delta^{l\top}) \\
&\approx \mathbb{E}(x^l x^{l\top}) \otimes \mathbb{E}(\delta^l \delta^{l\top}) = X^l \otimes \Delta^l.
\end{aligned}
\tag{6}
$$

Decomposing the Hessian matrix into components corresponding to different MLP layers makes it feasible to calculate its inverse for a large language model with an enormous number of parameters. To further release the memory pressure, we use the eigendecompositions of $X^l$ and $\Delta^l$:

$$
X^l = Q_{X^l} \Lambda_{X^l} Q_{X^l}^\top, \ \Delta^l = Q_{\Delta^l} \Lambda_{\Delta^l} Q_{\Delta^l}^\top.
\tag{7}
$$

By leveraging Eq. (6) and Eq. (7), we obtain a more accurate expression for $\mathbf{H}$:

$$
\begin{aligned}
\mathbf{H} &\approx (Q_X \otimes Q_\Delta) \Lambda (Q_X \otimes Q_\Delta)^\top, \\
\Lambda_{ii} &= \mathbb{E}[((Q_X \otimes Q_\Delta)D\theta)_i^2],
\end{aligned}
\tag{8}
$$

in which $\Lambda$ captures the variances of the pseudo-gradient projected onto each eigenvector of the K-FAC approximation. We then identify the 'important' MLP layers by choosing those with lower variances, as these layers exhibit enhanced stability. Reducing the number of calculation layers can significantly alleviate the storage pressure. However, it will also lead to a relatively small magnitude of the final result. To address the above issue, we introduce a dynamic scaling vector $\gamma$, which linearly scales the maximum absolute value in the calculated $\boldsymbol{\beta}$ to a predefined value $m$. We update the $\boldsymbol{\beta}$ values as shown in Eq. (9).

$$
\boldsymbol{\alpha} = \left.\frac{\partial \mathcal{Q}(\boldsymbol{\beta})}{\partial \boldsymbol{\beta}}\right|_{\boldsymbol{\beta}=(0,\ldots,0)}, \boldsymbol{\beta} = -\gamma \odot \boldsymbol{\alpha}, \gamma = \frac{m}{\max \text{abs}(\boldsymbol{\alpha})}.
\tag{9}
$$

### 3.4 IDEAL ALGORITHM

Our method maximizes the effectiveness of data and enhances the model's capabilities by adjusting the quantity ratio of the training set, thereby catalyzing the synergistic effect among data from different domains. Our approach leverages a trained model to assess and optimize the distribution of the training set. After calculating $\boldsymbol{\beta}$, we iteratively refine the training set composition, ultimately deriving an enhanced model from the foundational base model.

The IDEAL algorithm begins with an initial model $\mathcal{M}_0$ and a training set $\mathcal{D}_{tr}$ partitioned into domain-specific subsets $\{\mathcal{D}_1, \ldots, \mathcal{D}_n\}$. For

---

**Algorithm 1** IDEAL Algorithm

**Require:** Initial model $\mathcal{M}_0$, initial training set $\mathcal{D}_{tr} = \{\mathcal{D}_1, \ldots, \mathcal{D}_n\}$, maximum iterations $T$ (or stop criteria).
1: **for** $t = 1$ to $T$ **do**
2:     Train $\mathcal{M}_0$ on $\mathcal{D}_{tr,t}$ until optimal, resulting in $\mathcal{M}_t$;
3:     Test the performance of $\mathcal{M}_t$;
4:     Compute $\boldsymbol{\beta}$ following Eq. equation 9;
5:     Update domain-specific data: $\mathcal{D}_{i,t+1} \leftarrow (1 + \beta_i)\mathcal{D}_{i,t}$;
6:     Update training set: $\mathcal{D}_{tr,t+1} \leftarrow \sum_{i=1}^n \mathcal{D}_{i,t+1}$;
7:     **if** stopping criteria meet **then**
8:       **break**
9:     **end if**
10: **end for**

---

each iteration $t$, the model $\mathcal{M}_0$ is trained on the current training set $\mathcal{D}_{tr,t}$ until convergence, resulting in an updated model $\mathcal{M}_t$, as shown in Line 2. Next, the performance of $\mathcal{M}_t$ is evaluated on the reference dataset to assess its multi-task capabilities. Based on this evaluation, the parameter $\boldsymbol{\beta}$ is computed using the update rule defined in Eq. (9). In Line 5, the domain-specific data volumes are then adjusted proportionally, with each subset $\mathcal{D}_{i,t+1}$ updated as $(1 + \beta_i)\mathcal{D}_{i,t}$. We employ upsampling and downsampling to respectively increase or decrease the data volume of the dataset. The random sampling approach, on one hand, conserves computational resources and enhances processing speed, while simultaneously minimizing the potential impact of selection methods on the performance of IDEAL. The updated training set $\mathcal{D}_{tr,t+1}$ is reconstructed by aggregating the adjusted domain-specific subsets, as shown in Line 6. The process repeats until the stopping criteria are satisfied, ensuring an optimal and balanced dataset composition for training the final model. This iterative approach allows IDEAL to adaptively refine the dataset, enhancing model performance across diverse tasks.

## 4 EXPERIMENTS

### 4.1 EXPERIMENT SETUP

**Dataset Preparation.** We select four key domains: reasoning, mathematics, coding, and instruction-following to represent the model's diverse capabilities. These domains are among the most widely studied and representative areas of focus in the field, encapsulating critical aspects of model performance and alignment. In Section 4.2, we firstly evaluate models on BigBench Hard(BBH) Suzgun et al. (2022), GSM8K Cobbe et al. (2021), HumanEval Chen et al. (2021), and IFEval Zhou et al. (2023) benchmarks, which are widely used in previous works. In Section 4.3, we expand the evaluation benchmarks by adding MATH Hendrycks et al. (2021), ARC_C Clark et al. (2018), MBPP Austin et al. (2021b), and TruthfulQA Lin et al. (2021) benchmarks. Appendix A provides a detailed explanation of the benchmarks selected for our experiments, along with a comprehensive description of the specific preparation process for our training dataset.

**Training Setting:** We choose the LLama3.1-8B Grattafiori et al. (2024) as our base model $\mathcal{M}_0$ to adopt full fine-tuning. In all fine-tuning training experiments, we set the batch size to 256 and the maximum learning rate as $2 \times 10^{-5}$ with a cosine decay schedule. We train every model in 1 epoch and 3 epochs on $8 \times$ A100 GPUs and evaluate the result by using the OpenCompass platform Contributors (2023), respectively. The parameters mentioned above may not be optimal. However, we ensure that all our experiments use the same set of training parameters for consistency. Moreover, each experiment is repeated 5 times, and the average value is calculated to avoid the interference of random phenomena.

**Baseline.** We compare the performance of our IDEAL with other data training strategies as follows. (1) **Specific SFT**, which only uses a specific domain training data for SFT. (2) **Joint SFT**, where the different capability data directly combined. (3) **Random**, we randomly sample different data scales for each capability. (4) **DoReMi** Xie et al. (2024), which uses the group distributionally robust optimization (Group DRO) steps to generate new domain weights. (5) **DOGE** Fan et al. (2023), which determines the data proportions between domains by minimizing the discrepancy in backpropagation gradients. For our IDEAL, we set the parameter $m = 0.15$. In order to accelerate calculation speed, we sample the training set with a sample factor $\sigma = 0.5$.

### 4.2 MAIN RESULTS

In this section, we select individual benchmarks from four universal domains—mathematics, coding, reasoning, and instruction following—for evaluation, specifically GSM8K, HumanEval, BBH, and IFEval. Additionally, we configure two experimental settings with epochs set to 1 and 3, respectively. Training for 1 epoch is widely adopted due to its shorter training duration and lower risk of overfitting. Training for 3 epochs is aligned with practical applications and can achieve better SFT effects. These settings allow us to thoroughly assess the robustness and adaptability of our method across varied evaluation criteria and training durations. The main results are shown in the Table. 1.

Initially, we train four specific SFT models using the training sets corresponding to their respective capabilities. After domain-specific training, the base model demonstrates a $10\%$ improvement in

Table 1: Performance comparison of different baselines.

| Benchmark | | Mathematics | Coding | Reasoning | Instruction | Overall |
|---|---|---|---|---|---|---|
| Methods | Dataset | Acc(#Size) | Pass@1(#Size) | Average(#Size) | Average(#Size) | Average |
| **Epoch=1** | | | | | | |
| **Base Model** | - | 56.41( - ) | 27.44( - ) | 62.13( - ) | 12.22( - ) | 39.55 |
| **Specific** | Mathematics | 65.81(10.0k) | 0.00(0.0k) | 35.94(0.0k) | 22.54(0.0k) | 31.07 |
| | Coding | 48.14(0.0k) | 37.20(5.3k) | 2.99(0.0k) | 19.66(0.0k) | 27.00 |
| | Reasoning | 61.87(0.0k) | 7.32(0.0k) | 60.19(6.5k) | 26.70(0.0k) | 39.02 |
| | Instruction | 57.39(0.0k) | 46.95(0.0k) | 61.87(0.0k) | 22.47(2.0k) | 47.17 |
| **Joint** | $\mathcal{D}_0$ | 66.62(10.0k) | 41.26(5.3k) | 72.92(6.5k) | 38.36(2.0k) | 54.79 |
| **Random** | $\mathcal{D}_{random}^1$ | 63.84(3.5k) | 43.90(7.4k) | **75.11**(11.0k) | 39.70(3.0k) | 55.64 |
| | $\mathcal{D}_{random}^2$ | 63.23(1.7k) | 40.85(9.0k) | 74.56(12.0k) | 38.21(1.9k) | 54.21 |
| **DoReMi** | $\mathcal{D}_{1(DoReMi)}$ | 65.96(5.0k) | 41.63(8.0k) | 73.44(9.7k) | 34.26(1.0k) | 53.82 |
| | $\mathcal{D}_{2(DoReMi)}$ | 64.82(5.3k) | 43.90(12.0k) | 73.79(4.8k) | 38.16(1.5k) | 55.17 |
| **DOGE** | $\mathcal{D}_{1(DOGE)}$ | 64.82(11.5k) | 40.02(8.0k) | 74.99(3.2k) | 39.02(1.0k) | 54.71 |
| | $\mathcal{D}_{2(DOGE)}$ | 67.10(10.0k) | 42.24(12.0k) | 73.59(1.6k) | 30.53(0.5k) | 53.37 |
| **IDEAL** | $\mathcal{D}_{1(IDEAL)}$ | **68.01**(9.5k) | 44.51(6.0k) | 72.82(6.8k) | **39.78**(2.1k) | 56.28 |
| | $\mathcal{D}_{2(IDEAL)}$ | 67.55(9.0k) | **50.61**(7.1k) | 74.29(7.3k) | 39.03(1.9k) | **57.87** |
| **Epoch=3** | | | | | | |
| **Specific** | Mathematics | 74.45(10.0k) | 0.00(0.0k) | 30.82(0.0k) | 21.73(0.0k) | 31.75 |
| | Coding | 49.43(0.0k) | 41.46(5.3k) | 41.93(0.0k) | 28.74(0.0k) | 40.39 |
| | Reasoning | 62.09(0.0k) | 4.27(0.0k) | 65.69(6.5k) | 24.86(0.0k) | 39.23 |
| | Instruction | 58.91(0.0k) | 40.85(0.0k) | 60.90(0.0k) | 38.77(2.0k) | 49.86 |
| **Joint** | $\mathcal{D}_0$ | 68.66(10.0k) | 39.64(5.3k) | 74.11(6.5k) | 38.99(2.0k) | 55.35 |
| **Random** | $\mathcal{D}_{random}^1$ | 69.75(9.0k) | 45.12(7.5k) | 73.20(9.0k) | 35.81(3.0k) | 55.97 |
| | $\mathcal{D}_{random}^2$ | 64.82(1.7k) | 43.90(9.3k) | 70.60(5.0k) | 39.73(4.0k) | 54.76 |
| **DoReMi** | $\mathcal{D}_{1(DoReMi)}$ | 70.96(5.0k) | 40.85(8.0k) | 72.38(9.0k) | 36.00(1.5k) | 55.05 |
| | $\mathcal{D}_{2(DoReMi)}$ | 68.01(6.3k) | 44.51(10.0k) | 70.32(4.0k) | 42.16(3.0k) | 56.25 |
| **DOGE** | $\mathcal{D}_{1(DOGE)}$ | 70.17(11.0k) | 40.02(8.0k) | 73.44(4.0k) | 40.21(1.0k) | 55.96 |
| | $\mathcal{D}_{2(DOGE)}$ | 69.10(9.6k) | 42.24(12.0k) | 73.59(1.4k) | 42.53(1.0k) | 56.87 |
| **IDEAL** | $\mathcal{D}_{1(IDEAL)}$ | 70.17(10.0k) | 42.07(6.1k) | **74.71**(6.8k) | 45.25(2.1k) | 58.05 |
| | $\mathcal{D}_{2(IDEAL)}$ | **71.19**(11.0k) | **46.34**(7.0k) | 73.97(7.2k) | **45.40**(2.2k) | **59.23** |

performance or maintains its previously high scores. We further explain the specific SFT performance in Appendix E. By evaluating these specific SFT models, we validate the reliability of the quality of our training datasets. Subsequently, we integrate all four training sets into $\mathcal{D}_0$ and train a Joint SFT model, further enhancing its generalization capabilities across multiple domains. Based on Table 1, we analyze key findings that demonstrate our approach's effectiveness.

**Suboptimal initial data distribution.** Although the Joint SFT model demonstrates improved performance over the Specific SFT models across all datasets except HumanEval, we aim to avoid the "bucket effect" where the overall model performance is constrained by its weakest component. The random baselines also exhibit similar suboptimal performance. Although randomly determining the proportions of the four datasets could potentially yield different training outcomes, it lacks stability.

**IDEAL achieves optimal balance in 2 iterations.** By incrementally refining data ratios, IDEAL surpasses Joint SFT on all metrics and stabilizes performance across benchmarks, notably achieving HumanEval improvements(+9.35 improvement in the Epoch=1 setting) without compromising other tasks, fulfilling efficiency and stability requirements. We use bold and underline to denote the best and second-best performance scores, respectively. From the perspective of training data volume, IDEAL significantly increases the data amount in the Coding domain after iteration, while slightly reducing the data volume in the Mathematics and Instruction domains. Compared to DoReMi and DOGE, the changes in data volume by IDEAL are more stable. We argue that the key to the stable performance optimization of the model trained by IDEAL lies in searching for a better distribution around the initial distribution. Comparing the first and second rounds of the IDEAL method, the second-round model shows a slight decline in Mathematics and Instruction but achieves a substantial improvement in performance in the Coding domain. This demonstrates IDEAL's ability to directionally enhance model performance.

**Performance Degradation in HumanEval with Extended Training.** Although models trained for three epochs show improved performance across most domains, their capability on HumanEval is inferior to the optimal scores achieved with epoch=1. This trend is also observed in the results of

the Specific SFT models trained for three epochs, where the upper limit of scores on the HumanEval benchmark decreases as the number of training epochs increases. We infer that suboptimal training data distribution introduces conflicts among the data, and extended training exacerbates this issue, thereby inhibiting the model's ability to enhance performance in certain dimensions.

**Training different Epochs showcases different optimization priorities.** Training models over multiple epochs aims to leverage more data information and gradient updates. The IDEAL method controls the size of training sets across domains through the parameter $\beta$, where positive values of $\beta$ favor larger quantities, thereby providing more gradient information for that domain (even without introducing new data). In the epoch=3 setting, models are more likely to memorize existing information and thus require additional data and gradient updates. In Table 1, all four domains exhibit a demand for increased training data volume. In contrast, during the epoch=1 phase, IDEAL tends to locally adjust the data volume of each domain to achieve optimal performance. This selective adjustment highlights IDEAL's ability to fine-tune the dataset composition based on the model's training configuration, ensuring a more balanced and effective training process.

## 4.3 EXTENDED EXPERIMENT

We extend the TrustAI training set and double the number of benchmark evaluation criteria to ensure a more comprehensive and robust assessment of the model's performance. We also choose the Llama3.1 8B as our base model. To establish a clear upper bound on the performance achievable with the available data, we train domain-specific models using the entire domain-specific dataset, recorded as Specific SFT (FULL) in Table 5. We also introduce an additional experimental control group, $\mathcal{D}_{0(\text{FULL})}$, which includes all data from five training domains, comprising approximately 66k training samples, to investigate whether data volume plays a more critical role in model performance. We also conduct other extended experiments about model size and model architecture in Appendix D.

In this experiment, we adopt a balanced initial dataset $\mathcal{D}_0$, where 5k samples are randomly selected from each of the five domain-specific datasets to form a new dataset. Unlike the imbalanced domain distribution in Section 4.2, this balanced setup ensures a more equitable representation of each domain, making it easier to achieve consistent performance across tasks. Compared to $\mathcal{D}_{0(\text{FULL})}$, $\mathcal{D}_0$ contains fewer data and is a proper subset of it, allowing for a direct comparison to analyze the trade-off between data volume and distribution in multi-domain training. Additionally, this configuration presents a greater challenge for IDEAL, rigorously testing its robustness and adaptability in handling a uniformly distributed dataset.

Table 2: Results of the extended experiment

| Benchmark | | Mathematics | | Coding | | Reasoning | | Instruction | TrustAI | Overall |
|---|---|---|---|---|---|---|---|---|---|---|
| Methods | Dataset | GSM8K | MATH | HumanEval | MBPP | BBH | ARC_C | IFEval | Truthfulqa | Average |
| | | | | | Epoch=3 | | | | | |
| Base | - | 56.41 | 1.10 | 27.44 | 5.00 | 62.13 | 34.24 | 12.22 | 45.80 | 30.54 |
| Specific | Mathematics | 75.21 | 37.70 | 0.00 | 0.20 | 35.04 | 65.42 | 22.33 | 43.48 | 34.92 |
| | Coding | 6.75 | 4.74 | 40.24 | 43.00 | 51.00 | 33.22 | 35.15 | 25.65 | 29.97 |
| | Reasoning | 33.81 | 11.06 | 0.00 | 0.80 | 56.63 | 64.07 | 26.95 | 34.06 | 28.42 |
| | Instruction | 37.60 | 12.60 | 33.54 | 45.20 | 59.44 | 60.34 | 47.88 | 48.84 | 43.18 |
| | TrustAI | 16.00 | 8.74 | 7.32 | 48.60 | 26.50 | 70.85 | 55.95 | 17.06 | 31.38 |
| Joint | $\mathcal{D}_0$ | 69.83 | **30.94** | 42.68 | 44.20 | 69.32 | 72.88 | 47.20 | 55.22 | 54.03 |
| | $\mathcal{D}_{0(\text{FULL})}$ | 69.77 | 30.85 | 43.49 | 44.73 | 70.39 | 72.32 | 47.69 | 55.32 | 53.94 |
| IDEAL | $\mathcal{D}_{1(\text{IDEAL})}$ | 69.45 | 30.46 | 44.51 | 44.00 | 70.58 | 73.90 | **47.80** | 55.94 | 54.58 |
| | $\mathcal{D}_{2(\text{IDEAL})}$ | **70.51** | 30.20 | **44.51** | **45.60** | **72.12** | **74.92** | 47.78 | **55.94** | **55.20** |

**Data Volume Matters Little.** Comparing the results of Joint SFT and Joint SFT (FULL), we conclude that increasing data volume does not necessarily lead to improved model performance, even when the data quality is high. Past experience suggests that during the SFT phase, the volume of training data is not the critical factor; rather, the quality of the training data is the key determinant of model performance. Building on this, we add a further observation: in multi-domain joint training, data volume is not the sole concern; the distribution of training data across domains is equally as important as data quality in influencing the final model performance. This highlights the significance of balanced data allocation in achieving optimal results.

**IDEAL Demonstrates Robustness.** In both Section 4.2 and the experiment in this section, IDEAL consistently demonstrates a strong capability in optimizing data distribution, leading to improved

domain-specific scores for the model. Similarly, whether targeting individual capabilities or average performance, IDEAL achieves measurable enhancements within two iterations. However, as the number of evaluation targets increases, the reference set for validating average performance becomes more generalized, potentially losing the unique characteristics of individual datasets. This leads to less significant improvements in average performance; for instance, as shown in Table 5, IDEAL's average score increases by only 2.1%, compared to the 7% improvement observed in Section 4.2.

## 5 SENSITIVITY STUDY

In this section, we delve further into the hyperparameters and other factors that may influence the performance of IDEAL. In Section 5.1, we explore the optimal value of the parameter $m$. In Section 5.2, we clarify through experiments that the effectiveness of IDEAL is not solely due to changes in data volume.

### 5.1 SENSITIVITY TO THE SELECTION OF $m$ IN EQ. (9)

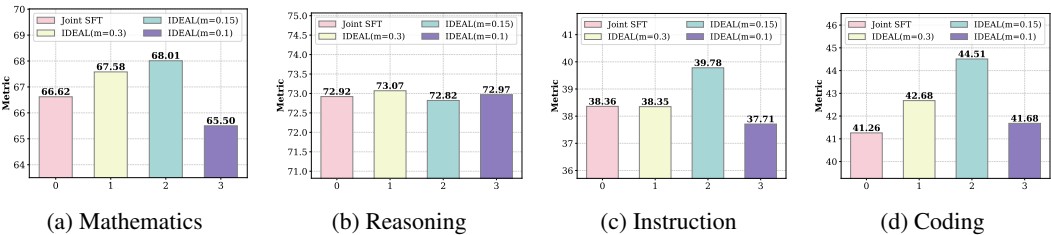

Figure 2: Models' performance on different domains with varying $m$ values.

As shown in Lemma 1, $\beta$ is essentially a small perturbation around 0. The upper bound of the scale $m$ plays a vital role in determining the magnitude of the adjustment of $\beta$. To explore the optimal range for $\beta$, we carry out experiments on three different settings for the choice of $m \in \{0.1, 0.15, 0.3\}$. Results on $\mathcal{D}_{1(\text{IDEAL})}$ in Sec. 4.2 after training 1 epoch are shown in Figure 2.

**The choice of $m$ should be neither large nor too small.** When $m = 0.1$, limited adjustments yield marginal gains due to insufficient data changes. Conversely, $m = 0.3$ causes unstable capability fluctuations as drastic shifts deviate from the original distribution. Based on our experiments, we recommend $m = 0.15$ as the optimal value, balancing moderate data adjustments and distribution integrity. This configuration achieves the highest average performance by enabling controlled yet impactful updates to data proportions.

### 5.2 SENSITIVITY TO THE VOLUME OF TRAINING DATA

Table 3: A comparison of performance across different baselines under the same data volume.

| Benchmark | | Mathematics | Coding | Reasoning | Instruction | Overall |
|---|---|---|---|---|---|---|
| Methods | Dataset | Acc(#Size) | Pass@1(#Size) | Average(#Size) | Average(#Size) | Average |
| Epoch=1 | | | | | | |
| DoReMi | $\mathcal{D}_{2(\text{DoReMi})}$ | 64.82(5.3k) | 43.90(12.0k) | 73.79(4.8k) | 38.16(1.5k) | 55.17 |
| | $\mathcal{D}_{2(\text{DoReMi'})}$ | 65.21(5.6k) | 43.90(12.8k) | 73.89(5.1k) | 38.45(1.6k) | 55.36 |
| DOGE | $\mathcal{D}_{2(\text{DOGE})}$ | 67.10(10.0k) | 42.24(12.0k) | 73.59(1.6k) | 30.53(0.5k) | 53.37 |
| | $\mathcal{D}_{2(\text{DOGE'})}$ | **67.73**(10.0k) | 42.78(12.8k) | 73.60(1.7k) | 31.12(0.5k) | 53.81 |
| IDEAL | $\mathcal{D}_{2(\text{IDEAL})}$ | 67.55(9.0k) | **50.61**(7.1k) | **74.29**(7.3k) | **39.03**(1.9k) | **57.87** |

The IDEAL method automatically adjusts the quantity of training data based on the calculated $\beta$ but does not control the total volume of training data. In the epoch=1 setting of Section 4.2, the total volume of training data in $\mathcal{D}_{2(\text{IDEAL})}$ is expanded compared to the reweighting methods of DoReMi and DOGE. To ensure a fair comparison, we apply the upsampling method to align the training data volume of DOGE and DoReMi with that of IDEAL and generate new training datasets $\mathcal{D}_{2(\text{DoReMi'})}$ and $\mathcal{D}_{2(\text{DOGE'})}$. We evaluate the models on these datasets and summarize results in Table 3.

**IDEAL outperforms other re-weighting methods not because of an increase in data volume**, but rather due to its ability to optimize the distribution of training data across domains more effectively and targeted. Although both DoReMi and DOGE exhibit improvements after upsampling (DoReMi increases from 55.17 to 55.36, and DOGE improves from 53.37 to 53.81), they still fall significantly short of IDEAL, which achieves an average score of 57.87. This highlights IDEAL's superior ability to optimize data distribution and improve model performance across diverse domains.

## 6 CONCLUSION

We propose a data equilibrium adaptation framework, IDEAL, which effectively optimizes dataset proportions for SFT. IDEAL is an iterative method that dynamically adjusts training data distributions and uses a simple and efficient manner, relying solely on existing datasets without the need for additional data or external resources. Experiments demonstrate IDEAL's robustness and effectiveness in diverse settings, consistently improving the multi-task performance of LLMs. Our framework provides a scalable and practical solution for preparing SFT training data, ensuring balanced and enhanced capabilities in LLMs for real-world applications.

## ETHICS STATEMENT

This study did not involve any unethical data collection, experimental procedures, or practices. All analyzes were conducted using publicly available or ethically obtained data, and no sensitive personal information was collected. The research design and methods adhere to established ethical standards for academic research, ensuring that no harm is caused to individuals, groups, or the environment.

## REPRODUCIBILITY STATEMENT

The paper fully discloses all the information needed to reproduce the main experimental results to the extent that it affects the main claims and/or conclusions. We release our code at `https://github.com/ming-bot/IDEAL` and our training dataset details are shown in Appendix A.

## ACKNOWLEDGMENT

This work is completed during Chenlin Ming's internship at the Shanghai Artificial Intelligence Laboratory and is supported by the Shanghai Artificial Intelligence Laboratory.

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

## A   DATASET AND TRAINING INFORMATION

**Reasoning:** We initially select BigBench Hard (BBH) Suzgun et al. (2022) as the benchmark to evaluate the reasoning capabilities of our model. BBH is a widely recognized benchmark designed to test a model's ability to handle complex and diverse reasoning tasks, making it an ideal choice for assessing our model's comprehensive reasoning skills. In section D, we further employ the challenge part of AI2's Reasoning Challenge (ARC) Clark et al. (2018) as our out-of-distribution evaluator. To further enhance the quality of the training dataset, we utilize the official BBH dataset as a foundation and employ GPT-4OpenAI et al. (2024) to regenerate the corresponding answers. This process allows us to refine and improve the quality of the dataset, ensuring that the training examples are both accurate and high-quality.

**Mathematics:** We select GSM8K Cobbe et al. (2021) as the benchmark to evaluate the mathematical reasoning capabilities of our model. GSM8K is a highly regarded dataset specifically designed to test models on a wide range of math problems, including arithmetic, algebra, and word problems. By covering diverse mathematical scenarios, GSM8K serves as a comprehensive tool for evaluating both the precision and depth of our model's mathematical understanding. In section D, we further add the MATH Hendrycks et al. (2021) benchmark. We start with the official GSM8K dataset and leverage the GPT-4 to generate the corresponding chain-of-thought (CoT) solutions. This approach allows us to refine the reasoning steps and enhance the clarity and accuracy of the solutions.

**Coding:** We choose HumanEval Chen et al. (2021) as the benchmark to assess the coding capabilities of our model. HumanEval is a well-known dataset specifically designed to evaluate a model's ability to understand, generate, and execute code. It provides a set of programming tasks that require not only syntactic correctness but also semantic understanding, logical reasoning, and problem-solving skills. We also add a new evaluator called the Mostly Basic Python problems (MBPP) Austin et al. (2021a) benchmark to more fairly evaluate the model's coding capability. However, due to the lack of an official training dataset for HumanEval, we construct a high-quality training set by sampling from the Tulu-Code dataset Ge et al. (2024), which contains 35k data.

**Instruction-following:** We select IFEval Zhou et al. (2023) as the benchmark to evaluate our model's instruction-following abilities. IFEval is designed to test a model's capacity to understand and execute diverse instructions, making it ideal for assessing alignment with user intent across various scenarios. Due to the limited size of the official IFEval training set, we enhance it by sampling additional data from WizardLM Evol-Instruct data Xu et al. (2023a). This combination creates a richer and more diverse training set, enabling our model to better generalize and excel in instruction-following tasks.

**TrustAI:** The reliability and truthfulness of large models are widely recognized as critical areas of focus. For evaluating model trustworthiness, we adopt TruthfulQA Lin et al. (2021) as our benchmark. Since TruthfulQA does not provide an open-source training dataset, we curated a new dataset by expanding the TruthGen Fulay et al. (2024) dataset, and used the GPT-4 to regenerate CoT answers.

**Reference Dataset:** The reference dataset serves as a validation set, independent from the training data, and is used to assess the model's generalization performance. Importantly, the reference dataset is strictly excluded from training to ensure unbiased evaluation. The performance of the model on this dataset gives us a comprehensive measure of the model's ability to generalize across all the tasks it was fine-tuned on. This is critical for validating that the IDEAL framework not only optimizes the training data distribution but also results in improved performance on unseen data. Our reference dataset consists of 500 representative samples selected from the training set, with any duplicates removed to ensure that the reference dataset is both unseen data, allowing for the evaluation of the model's generalization ability, and completely distinct from the test data, thus preventing any potential data leakage.

**Evaluation Metric:** For GSM8K, we adopt the 'accuracy' metric. For the HumanEval benchmark, we use the 'pass@1' metric to evaluate the probability that the code generated by the model in a single attempt successfully compiles. For the BBH benchmark, we consider the naive average metric to evaluate the average score of the model across multiple test capabilities in BBH. In IFEval, we adopt four metrics, namely prompt-level-strict-acc(P-s-acc), Inst-level-strict-acc(I-s-acc), prompt-level-loose-acc(P-l-acc), and Inst-level-loose-acc(I-l-acc), to comprehensively evaluate the model's

$$\frac{1}{N + \sum_{i=1}^{n} \beta_i |\mathcal{D}_i|} \arg\min_{\boldsymbol{\theta}} \left( \mathcal{L}(\mathcal{D}_{tr}, \boldsymbol{\theta}) + \sum_{i=1}^{n} \beta_i \mathcal{L}(\mathcal{D}_i, \boldsymbol{\theta}) + \lambda \|\boldsymbol{\theta} - \tilde{\boldsymbol{\theta}}\|^2 \right), \qquad (12)$$

capabilities in detail. The P-s-acc metric assesses the accuracy of the model's responses at the prompt level with strict criteria, while the I-s-acc evaluates the accuracy at the instance level with strict standards. On the other hand, the P-l-acc and I-l-acc use more relaxed criteria for evaluation at the prompt and instance levels, respectively. In our extended experiment, we use the 'accuracy' metric in MATH, ARC_C, and TruthfulQA. For MBPP, we adopt the 'score' metric to evaluate models' coding performance. After obtaining these values, we calculate their average to enable quick comparison among different models or experimental results, which provides a straightforward way to gauge the overall performance of the model in a more comprehensive manner.

## B    THEORETICAL ANALYSIS

### B.1    PROOF OF LEMMA 1

**Proof 1** *Since we assume in Eq. (1) that the model is trained to optimality on the training set $\mathcal{D}_{tr}$, we can obtain the following:*

$$\nabla \mathcal{L}(\mathcal{D}_{tr}, \theta^*) + \sum_{i=1}^{n} \beta_i \nabla \mathcal{L}(\mathcal{D}_i, \theta^*) = 0, \qquad (10)$$

*for any fixed $\beta$. Then calculate the derivative of both sides with respect to $\beta_j$. We obtain:*

$$\nabla^2 \mathcal{L}(\mathcal{D}_{tr}, \theta^*) \frac{\partial \theta^*}{\partial \beta_j} + \sum_{i=1}^{n} \beta_i \nabla^2 \mathcal{L}(\mathcal{D}_i, \theta^*) \frac{\partial \theta^*}{\partial \beta_j} \\ + \nabla \mathcal{L}(\mathcal{D}_j, \theta^*) = 0, \qquad (11)$$

*which implies Eq. (4).*

### B.2    GUARANTEE FOR INVERTIBLE HESSIAN

According to Assumption 1, we have $\nabla^2 \mathcal{L}$ as the Hessian matrix of the empirical risk function with respect to $\boldsymbol{\theta}$. When assuming $\mathcal{L}(\mathcal{D}, \boldsymbol{\theta})$ is strictly convex to $\boldsymbol{\theta}$, Hessian matrix $\nabla^2 \mathcal{L}$ is invertible. Now we consider the possibly-nonconvex case. When optimizing near a local minimum, the nonconvex problem can be locally approximated by a strictly convex optimization problem, allowing an invertible Hessian. In practice, once we obtain a parameter $\tilde{\boldsymbol{\theta}}$ by solving the inner problem Eq. equation 1, we can form a convex quadratic approximation of the objective around $\tilde{\boldsymbol{\theta}}$ by adding a damping term. Then problem Eq. (1) turns into Eq. (12).

where $\lambda > 0$. Now the invertible term in Eq. (5) changes to $\left( \nabla^2 \mathcal{L}(\mathcal{D}_{tr}, \boldsymbol{\theta}^*) + \lambda I \right)^{-1}$, providing a meaningful result.

## C    ESTIMATION ERROR ANALYSIS

In the process of implementing the IDEAL method, several factors may lead to estimation inaccuracies, which can potentially affect the overall performance and reliability of the proposed approach.

**Sub-optimality in Model Training.** The IDEAL method assumes that the model is trained to optimality on the training set $\mathcal{D}_{tr}$ as per Equation Eq. (1). However, in practical scenarios, to prevent overfitting, models are typically not trained to reach the globally optimal parameters. Instead, a balance is struck to obtain sub-optimal parameters that ensure good generalization across different domains. When the model is not trained to its full potential, the gradients and Hessian-related calculations used in our method, such as those in Lemma 1 for calculating $\frac{\partial \theta^*}{\partial \beta_m}$, may not accurately represent the true behavior of the model at its optimal state. This deviation from the ideal training

condition can introduce errors in the determination of the optimal mixing ratio $\beta$ for the training datasets.

**Methodological Errors from K-FAC for Hessian Matrix Computation.** To enable efficient calculation of the influence function, we rely on the K-FAC theory to decompose the Hessian matrix. As described in Section 3.3, we approximate the Hessian matrix $\mathbf{H}$ by decomposing it into a block-diagonal form according to different MLP layers. While this approximation significantly accelerates the inversion of the second-order gradient matrix, it inevitably introduces methodological errors. The block-diagonal approximation, where $\mathbf{H}^l \approx \mathbb{E}(x^l x^{l\top}) \otimes \mathbb{E}(\delta^l \delta^{l\top})$, simplifies the complex structure of the true Hessian matrix. This simplification means that the calculated influence function may deviate from the exact value. However, Martens Martens & Grosse (2015) provides a detailed analysis of estimation approximations and derives a theoretical upper bound for the approximation error in our Eq. equation 6:

$$|\kappa(x^l, x^{l\top}, \delta^l, \delta^{l\top})| + |\mathbb{E}(x^l)||\kappa(x^{l\top}, \delta^l, \delta^{l\top})| + |\mathbb{E}(x^{l\top})||\kappa(x^l, \delta^l, \delta^{l\top})|,$$

Here $\kappa(.)$ denotes the cumulant of its argument. Martens also includes extensive empirical validation of the estimation error. When inverting the approximated Hessian matrix to calculate $\frac{\partial \theta^*}{\partial \beta_m}$ in Eq. (4), these errors can propagate through the subsequent calculations of $\frac{\partial \mathcal{L}(\mathcal{D}_{ref}, \theta^*)}{\partial \beta_m}$ in Eq. (5).

**Experimental Errors due to Random Sampling for Accelerated Computation.** To expedite the computational process, we resort to random sampling from the training set. Although the law of large numbers assures that the mean of a large number of independent and identically distributed random samples converges to the expected value of the population, there is still a possibility of introducing random biases. In our method, when calculating expectations such as those in the decomposition of the Hessian matrix, the use of sampled data instead of the entire dataset can lead to errors. These random biases potentially result in inaccurate final results.

# D  SCALABILITY EXPERIMENT

## D.1  MODEL SCALABILITY

Table 4: Results of the model scalability(Qwen2.5 7B) experiment

| Benchmark | | Mathematics | | Coding | | Reasoning | | Instruction | TrustAI | Overall |
|---|---|---|---|---|---|---|---|---|---|---|
| Methods | Dataset | GSM8K | MATH | HumanEval | MBPP | BBH | ARC_C | IFEval | Truthfulqa | Average |
| Epoch=3 | | | | | | | | | | |
| Specific | Mathematics | 83.85 | 71.98 | 75.61 | 58.80 | 27.51 | 38.31 | 37.39 | 39.42 | 54.10 |
| | Coding | 82.41 | 65.74 | 78.05 | 2.20 | 69.17 | 40.00 | 52.88 | 60.00 | 56.30 |
| | Reasoning | 44.28 | 59.50 | 73.78 | 59.40 | 58.06 | 28.14 | 42.68 | 52.90 | 52.34 |
| | Instruction | 85.60 | 49.14 | 78.05 | 41.20 | 68.77 | 33.56 | 49.18 | 62.17 | 58.45 |
| Joint | $\mathcal{D}_0$ | 90.90 | 72.76 | 81.71 | 61.60 | 78.85 | 32.88 | 52.58 | 62.46 | 66.71 |
| IDEAL | $\mathcal{D}_{1(\text{IDEAL})}$ | 89.73 | 72.56 | 81.90 | 62.34 | 80.25 | 38.98 | 52.88 | 62.46 | 67.64 |
| | $\mathcal{D}_{2(\text{IDEAL})}$ | 91.20 | 72.76 | 81.71 | 62.18 | 81.93 | 44.27 | 55.47 | 61.90 | 68.93 |

To present IDEAL's universal on different model structures, we conduct a similar experiment in Table 5 on Qwen2.5 7B model. IDEAL achieved an average performance gain of +2.22 on Qwen2.5 7B, with a remarkable +11.39 improvement specifically on ARC-C, effectively addressing the limitations of simple Joint SFT training. These results collectively demonstrate IDEAL's cross-model generalization capability.

## D.2  MODEL SIZE SCALABILITY

To explore the adjustment scale to larger or smaller parameter regimes, we conduct extended experiments in Table 5 on smaller (Llama3.2 1B) and larger (Llama3.1 70B) variants. On a 1B model, IDEAL achieves a +0.77 average improvement over uniform mixing. Although the gain is lower than that observed on the 8B model (+1.26), we argue that this stems from the limited capacity of the 1B base model, which inherently constrains the effectiveness of IDEAL. On a 70B model, IDEAL still yields a +3.81 improvement, slightly outperforming the +1.26 gain on 8B. This confirms that our K-FAC implementation scale capability in practice.

Table 5: Results of the model size scalability experiment

| Benchmark | | Mathematics | | Coding | | Reasoning | | Instruction | TrustAI | Overall |
|---|---|---|---|---|---|---|---|---|---|---|
| Methods | Dataset | GSM8K | MATH | HumanEval | MBPP | BBH | ARC_C | IFEval | Truthfulqa | Average |
| Epoch=3 | | | | | | | | | | |
| Llama3.2 1B | $\mathcal{D}_0$ | 32.48 | 14.50 | 7.13 | 3.61 | 47.32 | 45.81 | 23.67 | 32.70 | 25.90 |
| | $\mathcal{D}_{1(\text{IDEAL})}$ | 32.67 | 14.82 | 6.93 | 3.70 | 46.91 | 44.78 | 25.32 | 33.00 | 26.02 |
| | $\mathcal{D}_{2(\text{IDEAL})}$ | 33.42 | 15.87 | 7.54 | 4.10 | 47.93 | 46.16 | 25.32 | 33.00 | 26.67 |
| Llama3.1 8B | $\mathcal{D}_0$ | 69.83 | 30.94 | 42.68 | 44.20 | 69.32 | 72.88 | 47.20 | 55.22 | 54.03 |
| | $\mathcal{D}_{1(\text{IDEAL})}$ | 69.45 | 30.46 | 44.51 | 44.00 | 70.58 | 73.90 | 47.80 | 55.94 | 54.58 |
| | $\mathcal{D}_{2(\text{IDEAL})}$ | 70.51 | 30.20 | 44.51 | 45.60 | 72.12 | 74.92 | 47.78 | 55.94 | 55.20 |
| Llama3.1 70B | $\mathcal{D}_0$ | 90.36 | 45.22 | 69.35 | 70.10 | 92.90 | 84.10 | 60.83 | 62.54 | 71.92 |
| | $\mathcal{D}_{1(\text{IDEAL})}$ | 90.36 | 47.61 | 72.13 | 72.60 | 92.71 | 86.77 | 65.13 | 61.98 | 73.66 |
| | $\mathcal{D}_{2(\text{IDEAL})}$ | 91.20 | 49.25 | 76.33 | 76.50 | 92.47 | 86.77 | 69.90 | 63.42 | 75.73 |

# E EXPLANATIONS OF DATA CONFLICTS AND DATA SYMBIOSIS

In the experiment conducted in Section 4.2, where the model is trained for one epoch, the performance of the Specific SFT models exhibit phenomena that warrant further explanation. In this section, we provide a detailed analysis of the results of the Specific SFT models trained for one epoch, as presented in Table 1, and offer pertinent interpretations of the observed phenomena.

**The Training set of instructions includes diverse data types.** In the context of Specific SFT, only the model trained on the instruction dataset demonstrates comprehensive capabilities, while other models exhibit significant performance degradation in specific domains. Upon examining the data content, we find that the selected training set included a variety of data types, such as mathematics, reasoning, coding, and more, to enhance the model's instruction-following ability. The high diversity of the data content enables the model to perform well across different domains. As a result, the trained model excels in all four selected domains and even outperformed models trained on domain-specific datasets in the areas of Coding and Reasoning. This phenomenon is further validated in subsequent experiments involving training for 3 epochs.

**Reasoning data enhances instruction-following and reasoning capabilities.** The Reasoning Specific SFT model trained on the BBH dataset demonstrates strong instruction-following capabilities. We hypothesize that the improvement in reasoning ability enables the model to better analyze user-provided instructions and engage in simple logical reasoning, thereby generating outputs that more closely align with expected results. The enhancement in reasoning ability also leads to improved performance in mathematics, which is intuitive given that mathematical reasoning can be viewed as a symbolic variant of logical reasoning. Mathematical proficiency still relies on the assistance of logical reasoning skills. This observation underscores the interconnectedness of reasoning and mathematical abilities in enhancing the model's overall instruction-following performance.

**Coding Data Appears Detrimental to Capabilities in Other Domains.** The model trained using the Coding dataset for SFT exhibits improved performance in the Coding domain, but its performance in other domains declines. We hypothesize that the exclusive focus on code-specific formats during training may cause the model to lose its ability to handle normal text-based dialogue data, thereby negatively impacting its performance in non-code-related domains. This suggests that while specialized training in coding enhances the model's coding proficiency, it may come at the cost of its generalizability and effectiveness in broader contexts.

Based on the phenomena and explanations discussed above, we can draw the following conclusions: **data from different domains exerts either positive or negative influences on capabilities in other domains, indicating the presence of Data Conflicts and Data Symbiosis**. When mixed datasets are used for training to achieve balanced improvements across multiple domains, the composition of the training dataset becomes a critical factor. This highlights the importance of carefully selecting and balancing data types to optimize the model's performance across diverse domains.

## F   SUPPLEMENTARY INFORMATION ABOUT $\beta$

In this section, we present the computed $\beta$ values calculated in Appendix D, along with the data composition distributions for both iterations of the IDEAL framework.

Table 6: $\beta$ values and training data distribution

| Benchmark | Dataset | Mathematics | Coding | Reasoning | Instruction | TrustAI |
|---|---|---|---|---|---|---|
| $\beta$ $(\times 10^{-5})$ | $\mathcal{D}_{1(\text{IDEAL})}$ | 7.34 | 84.30 | 15.74 | 18.02 | $-5.80$ |
| | $\mathcal{D}_{2(\text{IDEAL})}$ | 4.24 | 73.24 | 13.45 | 15.74 | $-4.17$ |
| **#Size** | $\mathcal{D}_{1(\text{IDEAL})}$ | 4.9k | 5.5k | 4.9k | 4.9k | 4.7k |
| | $\mathcal{D}_{2(\text{IDEAL})}$ | 4.9k | 6.3k | 5.0k | 5.1k | 4.7k |

The base model uses a uniform initial training data distribution, while IDEAL demonstrates stability and consistency in fine-tuning data quantities. The trends of changes across the five training domains remain consistent over two iterations and scale proportionally. As analyzed in Section 5.1, limiting the $\beta$ value ensures relatively stable model capabilities across all domains.

## G   SUPPLEMENTARY INFORMATION ABOUT WALL-CLOCK

In this section, we present our real running time of the scalability experiments in Appendix D. Since we employed an 8×A100 GPU configuration, the system can complete two full iterations of IDEAL training within 12 hours.

Table 7: Wall-clock in Appendix D

| Phases | EFLOPs | Actual |
|---|---|---|
| SFT-0 | 0.824 | 8h |
| IDEAL-1 | 3.164 | 32h |
| SFT-1 | 0.835 | 8h |
| IDEAL-2 | 3.159 | 32h |
| SFT-2 | 0.837 | 8h |
| Total | 8.819 | 88h |

## H   DECLARATION OF LLM USAGE

The paper only used LLMs for polishing grammar and language. No LLMs were involved in the design of the methodology, experimental process, data analysis, or any other core component of the research.

