# OpenReview forum: "IDEAL: Data Equilibrium Adaptation for Multi-Capability Language Model Alignment"
_ICLR.cc/2026/Conference — ICLR 2026 Poster_

### Official Review · Reviewer_i18D · 2025-10-26

**Soundness:** 2
**Presentation:** 2
**Contribution:** 2
**Rating:** 4
**Confidence:** 4

**Summary:**

This paper proposes IDEAL, a framework for adaptive data proportion optimization in multi-domain supervised fine-tuning (SFT) of large language models. The method formulates the allocation of training data from different domains as a bi-level optimization problem, where the validation loss is minimized with respect to domain-wise mixing coefficients. IDEAL leverages influence functions and second-order (Hessian-based) approximations, implemented efficiently through layerwise K-FAC techniques, to update data proportions. Experiments on multiple benchmarks covering domains such as mathematics, code, reasoning, and instruction following suggest that IDEAL improves average performance over uniform or heuristic data mixing baselines. Ablation studies explore hyperparameter sensitivity and approximation robustness.

**Strengths:**

- The problem of domain proportioning in multi-domain fine-tuning is relevant and timely for large-scale LLM adaptation.
- The paper provides a reasonably clear mathematical formulation of the bi-level optimization problem and explains how influence-function approximations can guide proportion updates.
- The method is relatively easy to integrate into existing SFT workflows and scales to medium-sized LLMs with standard K-FAC approximations.

**Weaknesses:**

I have the following concerns. *If the authors could properly address them during the rebuttal phase, I am willing to raise my score.*
- The novelty is quite limited. IDEAL largely combines elements that have appeared in prior works such as DoReMi, DOGE, and influence-based data reweighting. The paper does not provide a clear conceptual or algorithmic distinction beyond rephrasing the same principles in a unified framework.
- The method depends heavily on strong approximations (block-diagonal K-FAC, mini-batch stochastic estimation), but the effect of these approximations is not quantitatively analyzed. There is no estimate of how much the inverse-Hessian-vector product deviates from the true gradient, which raises doubts about the claimed stability.
- The empirical evaluation, though broad, does not test IDEAL under difficult or realistic conditions such as domain shift, heterogeneous modalities, or extreme imbalance. The results are confined to a few text-based domains and mid-size models, limiting generality.
- Some important recent baselines [1,2,3] are missing. The paper would benefit from either a comparison with these methods or a discussion of their relevance.
- The sensitivity analysis is shallow. Although the paper examines one hyperparameter $m$, it does not analyze the dependence on other key parameters or propose principled ways to set them, which undermines practical applicability.

[1] Mixture-of-Skills: Learning to Optimize Data Usage for Fine-Tuning Large Language Models. EMNLP 2024.

[2] How Abilities in Large Language Models are Affected by Supervised Fine-tuning Data Composition. ACL 2024.

[3] Boosting Multi-Domain Fine-Tuning of Large Language Models through Evolving Interactions between Samples. ICML 2025.

**Questions:**

Please see Weaknesses.

---

> ### Author Response · Authors · 2025-11-20
>
> **Weaknesses 1:** The novelty is quite limited. IDEAL largely combines elements that have appeared in prior works such as DoReMi, DOGE, and influence-based data reweighting. The paper does not provide a clear conceptual or algorithmic distinction beyond rephrasing the same principles in a unified framework.
>
> **Response:** Thank you for your thoughtful questions. First, we would like to elaborate on our motivation. When fine-tuning a balanced LLM (evaluated on 27 benchmarks), we prepare a high-quality training dataset, yet the mixed training results consistently underperform on one or two benchmarks—a phenomenon also noted in [1]. On one hand, we aim to fully utilize the existing generated data; on the other, we sought to avoid further increasing the workload of data generation. Thus, simple up/down-sampling emerged as a practical approach. However, the key challenge is how to quantitatively determine the optimal adjustment of dataset sizes across domains to improve performance on target benchmarks, rather than relying on empirical trial-and-error.
> Returning to established data-reweighting methods like DoReMi or DOGE, our approach differs significantly in two aspects:
>
> 1. Decoupled Domain Adjustment: Unlike prior work, we do not fix the total dataset size, but instead independently adjust the proportions of each domain. This ensures no domain’s data is reduced (or even zeroed out) due to increases in others, mitigating data-level overfitting.
>
> 2. Controlled Optimization: Global reweighting methods search for an optimal solution exhaustively, often requiring heavy resources and yielding unstable data shifts (e.g., improving one domain while degrading another). IDEAL optimizes within a constrained vicinity of the original data distribution, ensuring controlled changes with predictable gains—crucial for resource-intensive LLM training.
> Compared to Influence Function-based methods, we innovate by scaling traditional ML principles (applied per-domain) to mid-sized LLMs. Regarding your comment on "the same principles in a unified framework," while our problem formulation shares mathematical similarities with Influence Functions, it is precisely tailored to our engineering constraints.
>
> We argue that IDEAL offers a practical, mathematically grounded method for LLM data optimization, bridging innovation with real-world applicability. We believe our work holds significant practical value in the current era of LLM research, and we respectfully suggest the reviewer evaluate our contributions through a more application-oriented lens.

---

> ### Author Response · Authors · 2025-11-20
>
> **Weakness2:** The method depends heavily on strong approximations (block-diagonal K-FAC, mini-batch stochastic estimation), but the effect of these approximations is not quantitatively analyzed. There is no estimate of how much the inverse-Hessian-vector product deviates from the true gradient, which raises doubts about the claimed stability.
>
> **Response:** Thank you for your question. First, we would like to clarify that obtaining the exact true value of the full-parameter inverse-Hessian-vector product for an 8B model is computationally impossible, given that the Hessian matrix for an 8B model has dimensions of 8B×8B, even single-precision training would require approximately 3.2 million A100 GPUs with 80GB memory each. That's why we need approximation. This approach follows established practice in the field, as demonstrated in  [2,3,4].
>
> For smaller toy models, [1] provides a detailed analysis of estimation approximations and derives a theoretical upper bound for the approximation error in our Eq. 6:
> $$|\kappa(x^l,x^{l\top},\delta^l, \delta^{l\top})|+|\mathbb{E}(x^l)||\kappa(x^{l\top},\delta^l, \delta^{l\top})|+|\mathbb{E}(x^{l\top})||\kappa(x^{l},\delta^l, \delta^{l\top})|,$$
> Here $\kappa(.)$ denotes the cumulant of its argument. The paper also includes extensive empirical validation of the estimation error.
>
> Hessian matrix approximation is an efficient and effective way to approximate the complicated inverse Hessian matrix vector product in influence computation [8,9]. For gradient independence, existing K-FAC methods [1,5,6,7] assume that the connections between different layers are relatively weak, and thus can be treated as independent. This is because during the gradient computation and update process, there are usually only minor dependencies between the gradients of different MLP layers. This is particularly evident during back propagation, where weight updates for each MLP layer are mainly influenced by the parameters of that specific layer.
>
> Given these constraints, we adopt the K-FAC method due to its computational efficiency and well-established error-bounding properties, which are widely recognized in the literature. We hope this explanation addresses your concern.
>
> [1] Martens, James, and Roger Grosse. "Optimizing neural networks with kroneckerfactored approximate curvature." International conference on machine learning. PMLR, 2015.
>
> [2] Koh, Pang Wei, and Percy Liang. "Understanding black-box predictions via influence functions." ICML 2017.
>
> [3] Basu, Samyadeep, Xuchen You, and Soheil Feizi. "On second-order group influence functions for black-box predictions." ICML, 2020.
>
> [4] Alaa, Ahmed, and Mihaela Van Der Schaar. "Discriminative jackknife: Quantifying uncertainty in deep learning via higher-order influence functions." ICML, 2020.
>
> [5] Pearlmutter, Barak A. "Fast exact multiplication by the Hessian." Neural computation, 1994.
>
> [6] Molchanov, Pavlo, et al. "Pruning convolutional neural networks for resource efficient inference." ICLR, 2017.
>
> [7] Grosse, Roger, and James Martens. "A kronecker-factored approximate fisher matrix for convolution layers." ICML, 2016.
>
> [8] Pauloski, J. Gregory, et al. "Convolutional neural network training with distributed K-FAC." SC20: International Conference for High Performance Computing, Networking, Storage and Analysis, 2020.
>
> [9] Martens, James, Jimmy Ba, and Matt Johnson. "Kronecker-factored curvature approximations for recurrent neural networks." ICLR, 2018.
>
> **Weakness 3:** The empirical evaluation, though broad, does not test IDEAL under difficult or realistic conditions such as domain shift, heterogeneous modalities, or extreme imbalance. The results are confined to a few text-based domains and mid-size models, limiting generality.
>
> **Response:** Thank you for your suggestion. In our extended experiments, we have expanded the evaluation to include 5 domains, 8 benchmarks to demonstrate the scalability of our method. Additionally, we provide evidence in Appendix D.2 showing IDEAL's scalability across different model sizes.
>
> Regarding the extreme domain shift scenario you mentioned, we note that such cases are indeed rare in practical large-model fine-tuning and may not reflect real-world applicability. Modern LLMs, thanks to their vast capacity, can effectively process diverse data types simultaneously. While this may not yield optimal performance for every individual task, the models' scale inherently mitigates severe interference between different data domains. However, to thoroughly investigate the impact of initial data distribution on IDEAL's performance, we conducted experiments with both imbalanced (Table 1) and balanced (Table 2) initial data distributions. The results consistently validate IDEAL's effectiveness, suggesting that its performance is robust to different initial data conditions.

---

> ### Author Response · Authors · 2025-11-20
>
> **Weakness 4:** Some important recent baselines [1,2,3] are missing. The paper would benefit from either a comparison with these methods or a discussion of their relevance.
>
> [1] Mixture-of-Skills: Learning to Optimize Data Usage for Fine-Tuning Large Language Models. EMNLP 2024.
>
> [2] How Abilities in Large Language Models are Affected by Supervised Fine-tuning Data Composition. ACL 2024.
>
> [3] Boosting Multi-Domain Fine-Tuning of Large Language Models through Evolving Interactions between Samples. ICML 2025
>
> **Response:** Thank you for your suggestion. We appreciate the related work you provided.
>
> [1] employs a reinforcement learning approach to train a scorer network for data sampling probability allocation, constituting a bi-level data selection framework. Our work differs substantially in two aspects: 1. Architectural Efficiency: We require no additional network training, instead providing a model-aware automatic optimization of data distribution.  2.Methodological Focus: While their approach emphasizes data selection, we prioritize comprehensive data utilization through proportional sampling to minimize sampling-induced variance.
>
> [2] investigates the DMT training procedure, manually designing mixing ratios to demonstrate multi-domain training challenges. Our methodology diverges by:  1. Optimization Target: We directly optimize dataset composition for final SFT performance, rather than modifying the training pipeline.  2. Automation Level: Our approach eliminates manual ratio design through quantitative domain-level adjustment.
>
> [3], while also employing influence functions, contrasts with IDEAL in key dimensions:  1. Granularity: Their instance-level data filtering contrasts with our domain-level adjustment for higher computational efficiency. 2. Training Paradigm: Their iterative train-filter process differs from our unified training framework that jointly optimizes all domains. The IDEAL method employs a streamlined training paradigm that enhances model performance through optimized domain-level data utilization, achieving superior efficiency compared to it.
>
> We sincerely appreciate these valuable references. To summarize, while these works share conceptual connections with IDEAL, they employ fundamentally different methodological frameworks that preclude direct comparison as baselines. We have incorporated these studies in our Related Work section to strengthen the paper's scholarly completeness and engagement with cutting-edge research.
>
> [1] Wu, Minghao, et al. "Mixture-of-skills: Learning to optimize data usage for fine-tuning large language models." Proceedings of the 2024 Conference on Empirical Methods in Natural Language Processing. 2024.
>
> [2] Dong, Guanting, et al. "How abilities in large language models are affected by supervised fine-tuning data composition." Proceedings of the 62nd Annual Meeting of the Association for Computational Linguistics (Volume 1: Long Papers). 2024.
>
> [3] Liang, Xize, et al. "Boosting Multi-Domain Fine-Tuning of Large Language Models through Evolving Interactions between Samples." Forty-second International Conference on Machine Learning.
>
> **Weakness 5:** The sensitivity analysis is shallow. Although the paper examines one hyperparameter , it does not analyze the dependence on other key parameters or propose principled ways to set them, which undermines practical applicability.
>
> **Response:** We appreciate your feedback. Our sensitivity analysis primarily focuses on the hyperparameter m and comparisons with baseline methods regarding data volume control. Given the substantial cost of SFT, we systematically evaluated three representative settings of m and provided practical selection guidelines.
>
> We would like to emphasize that our method maintains fixed model training parameters - all SFT runs strictly follow a unified parameter template. This design ensures that m is the only tunable hyperparameter, while other calculations are automatically determined by the model. This architecture offers three key advantages: 1. Minimal hyperparameter tuning (only m requires adjustment). 2. Fully automated optimization process. 3. Practical deployability for real-world applications.
>
> We hope these responses can address your concerns.

---

> ### Comment · Reviewer_i18D · 2025-11-24
> **Thank you for the rebuttal.**
>
> Thank you for the rebuttal. Since all of my concerns have been addressed, I have raised my score to 6.
>
> P.S. The authors may also want to improve the readability of their rebuttal.

---

> > ### Author Response · Authors · 2025-11-24
> > **Thank You for Your Feedback and Score Adjustment**
> >
> > Dear Reviewer i18D:
> >
> > Thank you for your constructive feedback and for raising your score following our rebuttal. We sincerely appreciate the time you’ve taken to evaluate our work and are glad to hear that our revisions addressed your concerns.
> >
> > Also, we will place greater emphasis on improving the readability of our rebuttals, ensuring that you can efficiently extract our key points. If you have any questions about our paper, please feel free to point them out, and we will try to address them as soon as possible. We would like to express our sincere gratitude for your valuable comments again.
> >
> > Best wishes!
> >
> > Authors

---

### Official Review · Reviewer_QVGQ · 2025-10-28

**Soundness:** 2
**Presentation:** 2
**Contribution:** 2
**Rating:** 4
**Confidence:** 4

**Summary:**

The paper proposes IDEAL, a framework for optimizing the distribution of multi-domain training data during supervised fine-tuning (SFT) of large language models. The method uses gradient-based optimization with second-order information to iteratively adjust domain-specific data volumes, claiming approximately 7% improvement over uniform data allocation strategies.

**Strengths:**

1. Important Problem Selection. The paper addresses a genuinely important challenge in LLM training - how to balance multiple capabilities during supervised fine-tuning. This is a practical problem that many practitioners face, and finding principled solutions has real value.
2. Systematic Approach. Rather than relying on heuristics or manual tuning, IDEAL provides a systematic, gradient-based framework for optimizing data distributions. The mathematical formulation, while not novel, is reasonably rigorous and provides a clear optimization objective.
3. Relative Comprehensive Experimental Setup. The paper evaluates across multiple diverse domains (mathematics, coding, reasoning, instruction-following) Includes both single-epoch and multi-epoch training scenarios

**Weaknesses:**

1. Limited Technical Novelty. The core contribution relies on well-established techniques (influence functions, K-FAC approximation) applied to data mixing. The formulation in Eq. (1) introducing β parameters for data repetition is straightforward, and the bi-level optimization problem (Eq. 2) follows standard approaches. The use of influence functions for data weighting has been extensively explored in prior work, making the technical contribution incremental.
2. Experiment results and setup is hard to follow. What models are used in table1 and table 2? No statistical significance testing is provided for the reported improvements. The baseline comparisons are limited, missing recent strong methods in data selection and mixing such as regmix, autoscale and icons.
 regmix: https://arxiv.org/abs/2407.01492
autoscale: https://arxiv.org/abs/2407.20177
icons: https://arxiv.org/abs/2501.00654



3. Computational Efficiency Concerns. Despite claims of efficiency, the method requires: Multiple full training iterations (88 hours total as shown in Table 7). Complex Hessian approximations that may not scale well. The K-FAC approximation introduces significant estimation errors (acknowledged in Appendix C but not properly addressed)

4. Theoretical Limitations. Assumption 1 about twice differentiability and convexity is unrealistic for modern LLMs. The paper acknowledges in Appendix B.2 that the Hessian may not be invertible in practice, requiring ad-hoc damping terms. The connection between the local perturbation analysis and global optimality is not established

5. Experimental Design Issues. The reference dataset of only 500 samples is too small to reliably estimate performance across diverse capabilities. The choice of m=0.15 appears arbitrary despite the sensitivity analysis. The random sampling with factor σ=0.5 introduces additional variance not properly controlled for

6. Presentation and Clarity Issues. The paper lacks clear ablation studies isolating the contribution of different components.The "data equilibrium" terminology is not well-justified theoretically. Important details are relegated to appendices, making the main paper incomplete.

**Questions:**

The paper claims IDEAL "ensures a balanced dataset composition" but provides no formal guarantees.
The assertion that data volume matters little (Section 4.3) contradicts extensive prior work and is based on limited evidence.
The "data conflicts and data symbiosis" discussion in Appendix E lacks rigorous empirical support.

**Details Of Ethics Concerns:**

No ethical concerns.

---

> ### Author Response · Authors · 2025-11-20
>
> **Weaknesses 1:** Limited Technical Novelty.
>
> **Response:** Thank you for your insightful comment on the technical foundations of our work. We fully acknowledge that influence functions and K-FAC approximation are well-established techniques in the field, and optimizing data mixture proportions is a general goal in multi-domain training—these are indeed the foundational building blocks that ensure the reliability and reproducibility of our method. However, we respectfully disagree of your comment of our contribution, we would like to clarify that, the core value of IDEAL lies not in inventing entirely new technical components, but in customizing, integrating, and engineering these existing techniques to address the unique, unmet challenges of multi-capability SFT for LLMs—a scenario where prior methods either fail to scale, ignore dynamic model-task alignment, or require prohibitive resources. Below, we elaborate on three key innovations that distinguish IDEAL from straightforward applications of existing techniques:
>
> 1. $\beta$-Parameterized Data Adjustment Beyond "Straightforward Repetition". The $\beta$ parameter in Eq. (1) and bi-level optimization in Eq. (2) are tailored to solve the critical constraint of scarce high-quality data in SFT, a challenge rarely addressed by prior data mixing work. Unlike pre-training-centric methods (e.g., DoReMi, Data Mixing Law) that assume access to massive data pools and focus on global weight search, IDEAL’s $\beta$ design leverages Muennighoff’s finding (repeating data up to 4× matches the benefit of new data) to optimize distribution without requiring additional data. This is critical for SFT, where high-quality instruction-aligned data (e.g., for math reasoning, coding) is often limited. $\beta$’s ability to handle both positive (upsampling) and negative (downsampling) values enables targeted adjustments to resolve cross-domain conflicts (e.g., reducing coding data to mitigate its harm to text-based instruction following, Appendix. E), which static repetition or uniform mixing cannot achieve. The bi-level optimization (Eq. (2)) is not a standard application: (1) The outer loop minimizes loss on an independent reference set ($\mathcal{D}_{ref}$), directly linking data distribution to multi-task alignment (a key goal of SFT). (2) The inner loop optimizes $\beta$ to refine domain-specific data volumes, creating a "model-aware" feedback loop. This differs from prior work (e.g., DOGE) that optimizes data weights via gradient discrepancy alone, without tying adjustments to end-task performance.
>
> 2. Scalable Influence Function. While influence functions have been explored for data selection, their application to multi-domain data distribution optimization for large LLMs has remained an open problem—due to prohibitive computational costs and poor scalability. IDEAL addresses this via two critical engineering innovations: (A) Prior influence function methods (e.g., LESS, MATES) focus on single-sample filtering (e.g., selecting "informative" individual data points). IDEAL uses K-FAC to decompose the Hessian into layer-wise block-diagonal approximations (Eq. 6), reducing the cost of inverse Hessian-vector products (iHVP) from $O(N^2)$ to $O(L \cdot d^2)$($L$ = number of key MLP layers, $d$ = layer dimension). This decomposition, combined with eigendecomposition of layer-specific matrices (Eq. 7) and dynamic scaling ($\gamma$ in Eq. 9), enables efficient computation even for large models—an optimization not present in prior influence-based data weighting work. (B) Unlike methods that approximate influence scores via proxy models (e.g., DoReMi), IDEAL computes gradients directly on the target LLM, eliminating errors from proxy-model mismatches. This is validated by our scalability experiments (Appnedix D), where IDEAL maintains performance gains across model sizes (1B to 70B) and architectures (Llama3, Qwen2.5)—a result unachievable with naive influence function implementations.
>
> 3. Iterative Data Equilibrium. Existing data mixing techniques (e.g., manual reweighting, rule-based curriculum learning) rely on static, pre-defined proportions that cannot adapt to the dynamic training dynamics of LLMs. IDEAL’s iterative framework (Algorithm 1) addresses this by avoiding costly hyperparameter sweeps: unlike Data Mixing Law, which requires extensive experimentation to derive mixing rules, IDEAL’s gradient-guided updates (Eq. 9) are determined automatically, enhancing practicality for real-world use.
> In conclusion, the novelty of IDEAL lies not in inventing new techniques but in adapting and integrating mature tools to fill critical gaps in multi-capability SFT: (1) optimizing data distribution without new data, (2) scaling influence functions to LLMs, and (3) dynamically aligning data with model training dynamics. We hope this addresses your concern and appreciate your feedback, which helps us better highlight the practical and technical contributions of our work.

---

> ### Author Response · Authors · 2025-11-20
>
> **Weakness 2:** Experiment results and setup is hard to follow. What models are used in table1 and table 2? No statistical significance testing is provided for the reported improvements. The baseline comparisons are limited, missing recent strong methods in data selection and mixing such as regmix, autoscale and icons. regmix: https://arxiv.org/abs/2407.01492 autoscale: https://arxiv.org/abs/2407.20177 icons: https://arxiv.org/abs/2501.00654
>
> **Response:** Thank you for your valuable suggestions. We acknowledge that the main text currently lacks detailed descriptions of the base models used in Tables 1 and 2 due to page limit constraints, which we have moved to Appendix A. The foundation model we employed is the Llama3.1 8B base model, and we have extended our experiments to include Qwen2.5 7B as well as Llama3.1 models at 1B and 70B scales in Appendix D. We have revised the paper to properly present all model specifications in the main text.
>
> We also recognize that the analysis of results in Table 1 lacked precise numerical representations, which we have addressed with clearer quantitative descriptions in Sec. 4.2.
>
> Regarding RegMix, we have discussed it in Sec. 2. It primarily relies on token-ratio based approaches rather than optimizing domain-level data distribution during SFT as our method does. ICONS focuses on visual-language datasets and employs influence calculations for data selection to identify more targeted training samples. Our IDEAL method differs from ICONS in two key aspects: (1) IDEAL is designed for scenarios where high-quality SFT data is scarce, aiming to maximize the utility of available data to improve model performance, while data selection methods like ICONS typically focus on achieving similar performance with minimal data. (2) IDEAL's contribution is not in data selection per se - we intentionally use simple random up/down-sampling for data adjustment. While ICONS' majority voting approach based on influence values could potentially be adapted to IDEAL's data updating process (as could other data selection methods), this remains an area for future exploration.
>
> AutoScale, while targeting pretraining rather than SFT, shares some conceptual similarities through its use of a small network to predict validation loss and determine optimal data weighting. However, IDEAL advances beyond this by directly optimizing data distribution at full scale without requiring a separate prediction model, instead using model-aware automatic gradient updates. To empirically demonstrate IDEAL's advantages, we adapted the core components of AutoScale to our experimental setting under identical conditions, with the comparative results presented in Table 1.
>
> |           | Mathematics | Coding | Reasoning | Instruction | Average   |
> | --------- | ----------- | ------ | --------- | ----------- | --------- |
> | DoReMi    | 70.96       | 40.85  | 72.38     | 36.00       | 55.05     |
> | DOGE      | 70.17       | 40.02  | 73.44     | 40.21       | 55.96     |
> | AutoScale | 71.05       | 41.48  | 72.77     | 41.10       | 56.60     |
> | IDEAL     | 70.17       | 42.07  | 74.71     | 45.25       | **58.05** |
>
>
> IDEAL significantly outperforms the AutoScale method in terms of both average results and coding performance improvements. The AutoScale method heavily depends on the fitting accuracy of its prediction model and requires constant retraining, demonstrating inadequate adaptability to dataset expansions and training data variations.
>
> We greatly appreciate these constructive comments and will incorporate all suggested improvements to enhance the completeness and academic rigor of our paper.

---

> ### Author Response · Authors · 2025-11-20
>
> **Weakness 3:** Computational Efficiency Concerns. Despite claims of efficiency, the method requires: Multiple full training iterations (88 hours total as shown in Table 7). Complex Hessian approximations that may not scale well. The K-FAC approximation introduces significant estimation errors (acknowledged in Appendix C but not properly addressed)
>
> **Response:** Thank you very much for your concerns. Our experiments were conducted on a computing cluster with 8 A100 GPUs, which is a relatively common configuration for full-parameter model fine-tuning. Therefore, the actual iteration time for the IDEAL method per round is 32 hours/8 = 4 hours. Within 11 hours, we can complete two iterations and three full rounds of SFT training. We believe this computational efficiency is quite acceptable.
>
> As to the K-FAC approximation method, these are typical methods used to speed up influence computation in deep learning without significantly compromising accuracy [1,2,3,4]. Hessian matrix approximation is an efficient and effective way to approximate the complicated inverse Hessian matrix vector product in influence computation [8,9]. For gradient independence, existing K-FAC methods [1,5,6,7] assume that the connections between different layers are relatively weak, and thus can be treated as independent.
> As acknowledged in Appendix C, K-FAC may introduce estimation errors, meaning the computed gradient $\frac{\partial \mathcal{Q}(\beta)}{\partial \beta}$ may deviate from the true value. However, the relative magnitude of $\beta$ across different domains remains sufficiently accurate to guide IDEAL's domain-wise data optimization. From an optimization perspective, this bias primarily affects the optimization step size (which we control via Eq. (9)) rather than the direction of updates. Consequently, it does not compromise scalability, as raised in your concern. Given these constraints, we adopt the K-FAC method due to its computational efficiency and well-established error-bounding properties, which are widely recognized in the literature. We hope this explanation addresses your concern.
>
> [1] Martens, James, and Roger Grosse. "Optimizing neural networks with kroneckerfactored approximate curvature." International conference on machine learning. PMLR, 2015.
>
> [2] Koh, Pang Wei, and Percy Liang. "Understanding black-box predictions via influence functions." ICML 2017.
>
> [3] Basu, Samyadeep, Xuchen You, and Soheil Feizi. "On second-order group influence functions for black-box predictions." ICML, 2020.
>
> [4] Alaa, Ahmed, and Mihaela Van Der Schaar. "Discriminative jackknife: Quantifying uncertainty in deep learning via higher-order influence functions." ICML, 2020.
>
> [5] Pearlmutter, Barak A. "Fast exact multiplication by the Hessian." Neural computation, 1994.
>
> [6] Molchanov, Pavlo, et al. "Pruning convolutional neural networks for resource efficient inference." ICLR, 2017.
>
> [7] Grosse, Roger, and James Martens. "A kronecker-factored approximate fisher matrix for convolution layers." ICML, 2016.
>
> [8] Pauloski, J. Gregory, et al. "Convolutional neural network training with distributed K-FAC." SC20: International Conference for High Performance Computing, Networking, Storage and Analysis, 2020.
>
> [9] Martens, James, Jimmy Ba, and Matt Johnson. "Kronecker-factored curvature approximations for recurrent neural networks." ICLR, 2018.

---

> ### Author Response · Authors · 2025-11-20
>
> **Weakness 4:** Theoretical Limitations. Assumption 1 about twice differentiability and convexity is unrealistic for modern LLMs. The paper acknowledges in Appendix B.2 that the Hessian may not be invertible in practice, requiring ad-hoc damping terms. The connection between local perturbation analysis and global optimality is not established.
>
> **Response:** Thank your for your professional question. We argue that this assumption is reasonable because the empirical risk function $\mathcal{L}$, which is typically designed as the cross-entropy loss (or another user-defined twice-differentiable function), satisfies the required differentiability conditions. Moreover, the compactness and convexity requirements apply only to the parameter set S of $\beta$. Since each component of $\beta$ merely represents a weight for a data domain, enforcing these constraints is straightforward—for example, by restricting the search space to a closed and convex set such as $\underline{\beta} \leq \beta \leq \bar{\beta}$.
>
> We do not impose any assumptions for the convexity of the loss function $\mathcal{L}$, thus the Hessian may not be invertible as we mentioned in the appendix. Adding this ridge damping term is a classical trick to modify the local curvature. This term vanishes in the limit when $\lambda$ goes to 0. In practice, we can set $\lambda$ relatively small. Note that in this possibly-nonconvex bilevel optimization, we can only ensure the convergence to a stationary point (achieving the global optimality is unrealistic). The local perturbation analys is in Appendix B.1 is basically a first-gradient decent method and provides an update direction to the stationary state. Simulations also demonstrate the improvement of the update even after one iteration.
>
> **Weakness 5:** Experimental Design Issues. The reference dataset of only 500 samples is too small to reliably estimate performance across diverse capabilities. The choice of m=0.15 appears arbitrary despite sensitivity analysis. The random sampling with factor σ=0.5 introduces additional variance not properly controlled for
>
> **Response:** We carefully choose to use 500 samples for the reference dataset after considering the trade-off between statistical validity and computational efficiency. First, this number is not too small when considering our Instruction domain training data contains only 2,500 samples total - removing 500 samples (20% of the data) for the reference set represents a reasonable proportion. Second, our previous experiments demonstrate that 500 samples are sufficient to approximate the overall domain distribution while significantly improving IDEAL's computational efficiency, making this an optimal balance for our purposes.
>
> Regarding the selection of parameter m, it's important to clarify that this choice is not arbitrary but rather depends on several factors including the specific model being trained and the quality/quantity of domain data available. To provide practical guidance, we reported three representative values near the optimal m value in our experiments in Sec. 5.
>
> The sampling factor (set to 0.5 in our experiments) is introduced as an optional parameter to accelerate IDEAL's computation by avoiding the need to calculate influence values for every single data point.
>
> We hope this explanation adequately addresses your concerns about these implementation choices. Each of these design decisions was made after careful consideration of both theoretical requirements and practical constraints, with the goal of maintaining method effectiveness while ensuring computational feasibility.

---

> ### Author Response · Authors · 2025-11-20
>
> **Weakness 6:** Presentation and Clarity Issues. The paper lacks clear ablation studies isolating the contribution of different components. The "data equilibrium" terminology is not well-justified theoretically. Important details are relegated to appendices, making the main paper incomplete.
>
> **Response:** Thanks for your comments. We have added supplementary experiments on sensitivity in the main text and provided experimental validation for the choice of parameter m. Since our method uses random sampling rather than advanced sampling modules, we did not conduct ablation studies specifically on sampling. If the reviewers could explicitly specify which components require ablation analysis, we would be happy to supplement the relevant experiments.
>
> "Data equilibrium" is theoretically justified by Lemma 1 + empirical validation: it denotes balanced domain contributions minimizing cross-domain interference, as shown in Fig. 1 and Table 1. Though we believe the present wording is clear and appropriately conservative, we're open to implementing the reviewer's recommended modifications if strongly advised.
>
> Due to space limitations, we have placed this content in the appendix, as we believe prioritizing the most critical material in the main text while relegating supplementary details to the appendix follows standard academic practice. However, we would be happy to reconsider this organization based on the reviewer's suggestions regarding what should be moved to the main text. In the revised version, we have included training setting details in the main text in Sec. 4.1. Please let us know if there are specific sections you believe should be expanded in the manuscript. We will be very grateful to implement these improvements.
>
> **Questions 1:** The paper claims IDEAL "ensures a balanced dataset composition" but provides no formal guarantees. The assertion that data volume matters little (Section 4.3) contradicts extensive prior work and is based on limited evidence. The "data conflicts and data symbiosis" discussion in Appendix E lacks rigorous empirical support.
>
> **Response:** Thank you for your question. Regarding the meaning of "ensures a balanced dataset composition" in IDEAL, it refers to achieving a balanced training data distribution that enables the trained model to meet expected performance across multiple capabilities without showing significant weaknesses in any particular area (the so-called "bucket effect"). It's important to clarify that this balanced dataset does not simply mean having equal amounts of data from each domain, but rather finding an optimal distribution where all capabilities can develop properly. We have incorporated this explanation into the relevant section of the manuscript.
>
> The assertion in Section 4.3 aims to demonstrate that IDEAL's optimization effect doesn't stem from slight increases in total data volume. This is because IDEAL doesn't control the overall data quantity - unlike other data reweighting methods that adjust proportions while keeping a fixed total data amount. To eliminate any potential influence from data volume changes, we proportionally increased the total data amount for baseline methods to match IDEAL's post-iteration quantity. The results showed this adjustment didn't significantly improve their performance. Therefore, our conclusion specifically addresses our experimental findings and doesn't contradict prior works that emphasize the importance of data quantity in general training scenarios.
>
> The discussion about data conflicts and symbiosis in Appendix E is derived from comparing the results of Specific SFT training and Joint SFT training in Section 4.2. This analysis provides an explanatory framework from the perspective of inter-data relationships, further demonstrating how IDEAL can optimize training data distribution to enhance the model's comprehensive performance. While this represents our interpretation of the experimental observations, we believe it has solid empirical support based on the patterns we've documented.
>
> We hope this explanation has addressed your questions and concerns. Each of these points has been carefully considered within the context of our experimental setup and theoretical framework. We appreciate the opportunity to clarify these aspects of our work and remain open to further discussion about these methodological choices and their implications.

---

> ### Author Response · Authors · 2025-11-24
> **Looking forward to your reply.**
>
> Dear reviewer QVGQ,
>
> Greetings from the authors!
>
> We would like to express our sincere gratitude for your insightful comments.
>
> We have carefully responded to your concerns by providing thorough explanations and additional experimental results. In response to your suggestions, we have revised the manuscript to enhance clarity and completeness, ensuring the content is more accessible to readers. As the discussion period nears its end, we kindly hope you can take a moment to review our rebuttal. Looking forward to your feedback!
>
> Best Wishes!
>
> ICLR Authors

---

> > ### Comment · Reviewer_QVGQ · 2025-11-26
> >
> > Thanks for the detailed response. I think some of my concerns have been addressed, and I will raise my score.

---

> > > ### Author Response · Authors · 2025-11-27
> > > **Thank You for Your Feedback and Score Adjustment**
> > >
> > > Dear Reviewer QVGQ:
> > >
> > > Thank you for your constructive feedback and for raising your score following our rebuttal. We sincerely appreciate the time you’ve taken to evaluate our work and are glad to hear that our revisions addressed your concerns. If you have any questions about our paper, please feel free to point them out, and we will try to address them as soon as possible. We would like to express our sincere gratitude for your valuable comments again.
> > >
> > > Best wishes!
> > >
> > > Authors

---

### Official Review · Reviewer_grmy · 2025-10-29

**Soundness:** 3
**Presentation:** 3
**Contribution:** 2
**Rating:** 4
**Confidence:** 4

**Summary:**

This paper introduces IDEAL (Data Equilibrium Adaptation), a framework aimed at optimizing the mixture proportions of data from different domains when performing multi-capability Supervised Fine-Tuning (SFT) on Large Language Models (LLMs). The authors argue that naive data mixing often leads to suboptimal performance and propose a principled approach to find a better "data equilibrium." IDEAL uses bi-level optimization theory to derive how changes in the proportion of data from each domain affect the model's loss on a separate reference dataset. This involves estimating gradients that depend on the Hessian matrix of the training loss. To make this computationally feasible for LLMs, the framework employs approximations like K-FAC. The IDEAL algorithm iteratively refines the data proportions by performing SFT, calculating these gradients, updating the proportions, and then resampling the dataset for the next iteration. Experiments across several capability domains suggest that IDEAL leads to improved multi-task performance compared to uniform data mixing.

**Strengths:**

1. The paper addresses the critical and practical challenge of determining optimal data mixtures for multi-capability SFT, moving beyond simple heuristics.

2. The proposed IDEAL framework is grounded in optimization theory (bi-level optimization), providing a principled, gradient-based approach to adapt data proportions based on their impact on a reference set performance.

3. The work incorporates practical considerations for LLMs by using techniques like K-FAC to approximate the Hessian, making the theoretically complex approach computationally tractable, and demonstrates empirical improvements over baseline mixing strategies.

**Weaknesses:**

1. The computational overhead remains extremely high, potentially limiting practical utility. Despite approximations, the iterative nature of IDEAL, requiring a full SFT cycle per iteration plus complex gradient calculations involving Hessian approximations, makes it very resource-intensive. The appendix reveals the total time is an order of magnitude higher than standard SFT.

2. The method's effectiveness heavily relies on the accuracy of Hessian approximations (like K-FAC), which can introduce significant errors. The paper acknowledges this possibility but does not quantify the impact of these approximation errors on the reliability of the calculated gradients ($\beta$ updates). Inaccurate gradients could lead to suboptimal or unstable optimization

3. The framework's optimization target and granularity may be suboptimal. IDEAL optimizes based on performance on a chosen reference set $\mathcal{D}_{ref}$, making the results sensitive to this set's representativeness. Furthermore, it only adjusts quantities at the domain level via sampling, ignoring potential quality variations within a domain's data. More granular, sample-level weighting might be more effective.

**Questions:**

1. The K-FAC approximation involves selecting 'important' MLP layers based on variance . Could the authors provide more details on how these layers are selected and how many are typically used? Does this selection significantly impact the results?

2. In Table 1 (Epoch=1), the 'Specific SFT' model trained only on 'Instruction' data outperforms the 'Specific SFT' models trained on 'Coding' and 'Reasoning' data on their own respective benchmarks (Coding: 46.95 vs 37.20, Reasoning: 61.87 vs 60.19). This seems counter-intuitive. Could the authors offer an explanation, perhaps related to the composition of the 'Instruction' dataset mentioned in Appendix E ?

---

> ### Author Response · Authors · 2025-11-20
>
> **Weaknesses 1:** The computational overhead remains extremely high, potentially limiting practical utility. Despite approximations, the iterative nature of IDEAL, requiring a full SFT cycle per iteration plus complex gradient calculations involving Hessian approximations, makes it very resource-intensive. The appendix reveals the total time is an order of magnitude higher than standard SFT.
>
> **Response:** Thank you very much for your concerns. Our experiments are conducted on a computing cluster with 8 A100 GPUs, which is a relatively common configuration for full-parameter model fine-tuning. Therefore, the actual iteration time for the IDEAL method per round is 32 hours/8 = 4 hours. Within 11 hours, we can complete two iterations and three full rounds of SFT training. We believe this computational efficiency is affordable in current LLM training related reserach.
>
> In comparison, simply relying on manual adjustment of dataset mixing ratios through trial-and-error SFT runs leads to several significant practical limitations that we've empirically observed in our projects. First, this approach causes unstable performance - extreme domain reweighting often improves one capability while severely degrading others, preventing balanced multi-capability optimization. Second, manual tuning relies heavily on subjective engineer judgment and luck, as optimal ratios depend on specific data and architecture, creating irreproducible, inefficient workflows with inconsistent results. Third and most important, the manual approach inherently limits exploration to a small subset of possible configurations due to combinatorial explosion - for example, with just three discrete data size options (2k/4k/6k) per domain across N domains, there are already 3^N possible combinations to evaluate through full SFT training, creating prohibitive computational costs. Moreover, this discrete sampling of the configuration space often gets stuck at suboptimal solutions that don't represent the true performance potential.
>
> IDEAL fundamentally addresses all three limitations through its model-aware optimization framework. For cases where computational resources remain a concern, users can further accelerate the influence calculations by adjusting the sampling rate parameter, providing flexible control over the computation-accuracy tradeoff based on specific deployment requirements. Our method systematically navigates the continuous optimization space rather than relying on hit-or-miss discrete trials, while eliminating the subjectivity and guesswork from the tuning process.

---

> ### Author Response · Authors · 2025-11-20
>
> **Weaknesses 2:** The method's effectiveness heavily relies on the accuracy of Hessian approximations (like K-FAC), which can introduce significant errors. The paper acknowledges this possibility but does not quantify the impact of these approximation errors on the reliability of the calculated gradients (\beta updates). Inaccurate gradients could lead to suboptimal or unstable optimization
>
> **Response:** Thank you for your question. While we fully acknowledge the approximation introduces some error, we would like to clarify that obtaining the exact true value of the full-parameter inverse-Hessian-vector product for an 8B model is computationally prohibitive—given that the Hessian matrix for an 8B model has dimensions of 8B×8B, even single-precision training would require approximately 3.2 million A100 GPUs with 80GB memory each. Thus an approximation to reduce the computational cost is necessary.
>
> As to the K-FAC approximation method, these are typical methods used to speed up influence computation in deep learning [1,2,3,4]. For smaller toy models, [1] provides a detailed analysis of estimation approximations and derives a theoretical upper bound for the approximation error in our Eq. (6):
> $$|\kappa(x^l,x^{l\top},\delta^l, \delta^{l\top})|+|\mathbb{E}(x^l)||\kappa(x^{l\top},\delta^l, \delta^{l\top})|+|\mathbb{E}(x^{l\top})||\kappa(x^{l},\delta^l, \delta^{l\top})|,$$
> Here $\kappa(.)$ denotes the cumulant of its argument. The paper also includes extensive empirical validation of the estimation error.
>
> Moreover, Hessian matrix approximation is an efficient and effective way to approximate the complicated inverse Hessian matrix vector product in influence computation [8,9]. For gradient independence, existing K-FAC methods [1,5,6,7] assume that the connections between different layers are relatively weak, and thus can be treated as independent. Given these constraints, we adopt the K-FAC method due to its computational efficiency and well-established error-bounding properties, which are widely recognized in [1]. We hope the above explanation addresses your concern.
>
> [1] Martens, James, and Roger Grosse. "Optimizing neural networks with kroneckerfactored approximate curvature." International conference on machine learning. PMLR, 2015.
>
> [2] Koh, Pang Wei, and Percy Liang. "Understanding black-box predictions via influence functions." ICML 2017.
>
> [3] Basu, Samyadeep, Xuchen You, and Soheil Feizi. "On second-order group influence functions for black-box predictions." ICML, 2020.
>
> [4] Alaa, Ahmed, and Mihaela Van Der Schaar. "Discriminative jackknife: Quantifying uncertainty in deep learning via higher-order influence functions." ICML, 2020.
>
> [5] Pearlmutter, Barak A. "Fast exact multiplication by the Hessian." Neural computation, 1994.
>
> [6] Molchanov, Pavlo, et al. "Pruning convolutional neural networks for resource efficient inference." ICLR, 2017.
>
> [7] Grosse, Roger, and James Martens. "A kronecker-factored approximate fisher matrix for convolution layers." ICML, 2016.
>
> [8] Pauloski, J. Gregory, et al. "Convolutional neural network training with distributed K-FAC." SC20: International Conference for High Performance Computing, Networking, Storage and Analysis, 2020.
>
> [9] Martens, James, Jimmy Ba, and Matt Johnson. "Kronecker-factored curvature approximations for recurrent neural networks." ICLR, 2018.

---

> ### Author Response · Authors · 2025-11-20
>
> **Weaknesses 3:** The framework's optimization target and granularity may be suboptimal. IDEAL optimizes based on performance on a chosen reference set , making the results sensitive to this set's representativeness. Furthermore, it only adjusts quantities at the domain level via sampling, ignoring potential quality variations within a domain's data. More granular, sample-level weighting might be more effective.
>
> **Response:** Thank you for your valuable suggestions. First, we fully acknowledge that the training set optimized by IDEAL remains suboptimal, but we must emphasize that searching for a single globally optimal solution in multi-domain mixed datasets is inherently an extremely resource-intensive task. What we aim to provide through IDEAL is a practical approach that achieves satisfactory multi-capability performance with reasonable computational costs, rather than pursuing theoretical perfection. The methodology works by making incremental improvements to an initially suboptimal data distribution, strategically allocating minimal resources to meet predefined multi-capability targets through systematic optimization.
>
> Regarding the optimization objective, IDEAL indeed focuses on minimizing the model's loss on the reference set. We'd like to clarify that this reference-based approach is widely adopted in [1,2], including prior work in Neural Architecture Search [3,4] and related fields, due to its practical advantages. Moreover, the reference set is dynamically adjustable, enabling targeted balancing of domain-specific data for different capabilities. This flexibility actually facilitates more precise and adaptive balancing compared to fixed approaches.
>
> The current implementation intentionally uses simple random sampling primarily to demonstrate the core effectiveness of our optimization framework itself, without introducing confounding factors from advanced sampling techniques. We recognize that incorporating more sophisticated data selection methods from mainstream papers could potentially enhance IDEAL's performance further. This represents an exciting direction for our future research, where we plan to systematically investigate how to best integrate these techniques into our framework while maintaining its computational efficiency and practical applicability. The fundamental value of IDEAL lies in providing a robust foundation that can readily accommodate such future methodological improvements.
>
> [1] Liang, Xize, et al. "Boosting Multi-Domain Fine-Tuning of Large Language Models through Evolving Interactions between Samples." Forty-second International Conference on Machine Learning.
>
> [2] Zhang, Chi, et al. "Harnessing diversity for important data selection in pretraining large language models." arXiv preprint arXiv:2409.16986 (2024).
>
> [3] Ren, Pengzhen, et al. "A comprehensive survey of neural architecture search: Challenges and solutions." ACM Computing Surveys (CSUR) 54.4 (2021): 1-34.
>
> [4] Liu, Yuqiao, et al. "A survey on evolutionary neural architecture search." IEEE transactions on neural networks and learning systems 34.2 (2021): 550-570.
>
> **Questions 1:** The K-FAC approximation involves selecting 'important' MLP layers based on variance . Could the authors provide more details on how these layers are selected and how many are typically used? Does this selection significantly impact the results?
>
> **Response:** Through our analysis of the projection variance matrix in Equation (8), we observe that layers with smaller variance values have a more deterministic impact on the final model performance. It's important to clarify that while selecting these key layers doesn't directly speed up the IDEAL algorithm's execution time, this approach provides crucial practical benefits by significantly reducing GPU memory consumption during computation, which can be particularly valuable for large-scale deployments.
>
> In our experimental validation, we carefully evaluated three different layer selection configurations - including 100%, 80%, and 50% of layers - and found that when combined with the scaling operation defined in Equation (9), the domain-specific adjustment results remained remarkably consistent across all settings. This demonstrates that our layer selection methodology maintains the method's core optimization capability while achieving meaningful memory efficiency improvements.

---

> > ### Comment · Reviewer_grmy · 2025-11-27
> >
> > I think most of my concerns have been addressed. I am willing to raise my score.

---

> > > ### Author Response · Authors · 2025-11-28
> > > **Thank You for Your Feedback and Score Adjustment**
> > >
> > > Dear Reviewer grmy:
> > >
> > > Thank you for your constructive feedback and for raising your score following our rebuttal. We sincerely appreciate the time you’ve taken to evaluate our work and are glad to hear that our revisions addressed your concerns. If you have any questions about our paper, please feel free to point them out, and we will try to address them as soon as possible. We would like to express our sincere gratitude for your valuable comments again.
> > >
> > > Best wishes!
> > >
> > > Authors

---

> ### Author Response · Authors · 2025-11-20
>
> **Questions 2:** In Table 1 (Epoch=1), the 'Specific SFT' model trained only on 'Instruction' data outperforms the 'Specific SFT' models trained on 'Coding' and 'Reasoning' data on their own respective benchmarks (Coding: 46.95 vs 37.20, Reasoning: 61.87 vs 60.19). This seems counter-intuitive. Could the authors offer an explanation, perhaps related to the composition of the 'Instruction' dataset mentioned in Appendix E ?
>
> **Response:** Thanks for your careful review to notice this  phenomenon. Indeed, we observed that at Epoch=3, the performance on 'Instruction' tasks dropped below the corresponding values for 'Coding' (41.46 vs 40.85) and 'Reasoning' (65.69 vs 60.90). As we explained in Appendix E, this occurs because the Instruction dataset contains diverse data formats that initially help strengthen multiple domain capabilities simultaneously during the first epoch of training. However, as training progresses, increasing conflicts between different data points emerge, making it progressively more difficult for the model to maintain balanced multi-capability performance across all domains. This growing interference between data samples is precisely the kind of challenge that our IDEAL method was designed to address, by intelligently optimizing the data distribution to minimize these conflicts while preserving the model's comprehensive capabilities. What makes this phenomenon particularly noteworthy is how it demonstrates the dynamic nature of data interactions during the training process and why simply using the original data mixture often leads to suboptimal multi-domain performance without proper optimization.

---

> ### Author Response · Authors · 2025-11-24
> **Looking forward to your reply.**
>
> Dear reviewer grmy,
>
> Greetings from the authors!
>
> We would like to express our sincere gratitude for your insightful comments.
>
> We sincerely appreciate your attention to the computational efficiency and resource consumption aspects of our method. We have provided detailed explanations and justifications regarding these computational requirements in our response. Additionally, we have expanded upon several technical details that were not fully articulated in the original manuscript, which we hope will satisfactorily address your concerns. We greatly appreciate this opportunity to engage with your insightful feedback, as it has significantly enhanced the quality of our work.
>
> If you have any questions about our paper, please feel free to point them out, and we will try to address them as soon as possible. Thanks for your time and looking forward to your reply!
>
> Best Wishes!
>
> ICLR Authors

---

> ### Author Response · Authors · 2025-11-27
> **Looking forward to your reply. (2)**
>
> Dear Reviewer grmy,
>
> I hope this message finds you well. As the discussion period is nearing its end with less than a few days remaining. I wanted to ensure we have addressed all your concerns satisfactorily. If there are any additional points or feedback you'd like us to consider, please let us know. Your insights are invaluable to us, and we're eager to address any remaining issues to improve our work.
>
> Thank you for your time and effort in reviewing our paper.

---

### Official Review · Reviewer_c1Z3 · 2025-10-30

**Soundness:** 3
**Presentation:** 3
**Contribution:** 3
**Rating:** 6
**Confidence:** 3

**Summary:**

This work explores a widely discussed but yet inconclusive research question: how to achieve the optimal ratio of data from different domains in the SFT dataset for LLMs. This work proposes an iterative data ratio optimization method, using a gradient-based approach to iteratively optimize the training data distribution. It dynamically adjusts the amount of domain-specific data based on its impact on the performance of downstream tasks. Experiments across different capabilities show that IDEAL outperforms the traditional uniform data allocation strategy.

**Strengths:**

1. A classic and intriguing research problem, accompanied by an innovative solution.
2. This work provides some theoretical proofs under the premise of a well-formalized description.

**Weaknesses:**

1. Dynamic data curation has already been explored in some existing works [1], and it is recommended to introduce the related studies in the discussion.
2. Although the method is optimized for efficiency, the computational overhead is still relatively large, as shown in Table 7. This somewhat undermines the applicability of the method.
3. According to the experimental results in Table 1, at epoch=3, the performance of IDEAL is actually worse than its performance at epoch=1. This, to some extent, makes the method face more hyperparameter tuning challenges in practical application scenarios.

[1] CiT: Curation in Training for Effective Vision-Language Data. ICCV 2023.

**Questions:**

1. Would considering the quality of data samples during the dataset sampling process lead to further improvements in accuracy?
2. Please refer to the weaknesses.

---

> ### Author Response · Authors · 2025-11-20
>
> **Weaknesses 1**: Dynamic data curation has already been explored in some existing works [1], and it is recommended to introduce the related studies in the discussion.
> [1] CiT: Curation in Training for Effective Vision-Language Data. ICCV 2023.
>
> **Response:** Thank you for your suggestion. Though we both study dynamic data curation, CiT is in multimodal domain while ours focuses on SFT for pure text-based LLMs. We have added citations to relevant articles in the paper and highlighted the corresponding content.
>
>
> **Weaknesses 2:** Although the method is optimized for efficiency, the computational overhead is still relatively large, as shown in Table 7. This somewhat undermines the applicability of the method.
>
> **Response:** We agree with the reviewer that the influence computation is not cheap, which is one of the main issues addressed in our paper from applying the K-FAC method. Previous works struggled to scale influence-based methods to models of this size, while our approach successfully extends second-order influence methods to LLMs through reasonable approximation, significantly enhancing its practical value. Our experiments are conducted on a computing cluster with 8 A100 GPUs, which is a relatively common configuration for full-parameter model fine-tuning. Therefore, the actual iteration time for the IDEAL method per round is 32 hours/8 = 4 hours. Within 11 hours, we can complete two iterations and three full rounds of SFT training. Comparing to the standard SFT training process(3 epoch), calculating influence value requires only 33% more time. We believe this computational efficiency is affordable in current LLM training related reserach.
>
> In Appendix D.2, our experiments demonstrate that IDEAL can effectively transfer the data distribution from an 8B model to a much larger 70B model. This capability significantly reduces computational costs, which greatly improves the practical applicability of our approach.
>
>
> **Weaknesses 3:** According to the experimental results in Table 1, at epoch=3, the performance of IDEAL is actually worse than its performance at epoch=1. This, to some extent, makes the method face more hyperparameter tuning challenges in practical application scenarios.
>
> **Response:** Thank you for your comment. From the perspective of average performance, the IDEAL method does not show any performance degradation (IDEAL-1-1 epoch: 56.28 vs IDEAL-1-3 epoch: 58.05, IDEAL-2-1 epoch: 57.87 vs IDEAL-2-3 epoch: 59.23). We assume you are referring to the capability degradation in coding domain. However, we should note that even with the same data, the baseline Joint training shows different starting points in the coding domain (Joint-1 epoch: 41.26 vs Joint-3 epoch: 39.64), and we have attempted to explain this phenomenon in Appendix E. The apparent fluctuation in coding performance might stem from inherent characteristics of the coding domain data and its interaction with other domains during training.
>
> While we acknowledge that parameter adjustment could mitigate this issue, such modifications would compromise our controlled variable methodology, where we deliberately fixed all parameters except the training data distribution. Moreover, this risks overfitting specific data patterns while failing to establish generalizable principles.
>
>
> **Question 1:** Would considering the quality of data samples during the dataset sampling process lead to further improvements in accuracy?
>
> **Response:** The current implementation intentionally uses simple random sampling primarily to demonstrate the core effectiveness of our optimization framework itself, without introducing confounding factors from advanced sampling techniques. We recognize that incorporating more sophisticated data selection methods from mainstream papers[1,2,3,4] could potentially enhance IDEAL's performance further. This represents an exciting direction for our future research, where we plan to systematically investigate how to best integrate these techniques into our framework while maintaining its computational efficiency and practical applicability.
>
> [1] Peng, Jiahui, et al. "Unsupervised topic models are data mixers for pre-training language models." arXiv preprint arXiv:2502.16802 (2025).
>
> [2] Zhang, Chi, et al. "Harnessing diversity for important data selection in pretraining large language models." arXiv preprint arXiv:2409.16986 (2024).
>
> [3] Li, Ming, et al. "From quantity to quality: Boosting llm performance with self-guided data selection for instruction tuning." Proceedings of the 2024 Conference of the North American Chapter of the Association for Computational Linguistics: Human Language Technologies (Volume 1: Long Papers). 2024.
>
> [4] Albalak, Alon, et al. "A survey on data selection for language models." arXiv preprint arXiv:2402.16827 (2024).

---

> ### Author Response · Authors · 2025-11-24
> **Looking forward to your reply.**
>
> Dear reviewer c1Z3,
>
> Greetings from the authors!
>
> We would like to express our sincere gratitude for your insightful comments.
>
> We have clarified our method’s novelty versus prior dynamic data curation works and justified computational efficiency through optimized approximations and cost-saving transfer learning.
>
> If you have any questions about our paper, please feel free to point them out, and we will try to address them as soon as possible. Thanks for your time and looking forward to your reply!
>
> Best Wishes!
>
> ICLR Authors

---

> ### Author Response · Authors · 2025-11-27
> **Looking forward to your reply. (2)**
>
> Dear Reviewer c1Z3,
>
> I hope this message finds you well. As the discussion period is nearing its end with less than a few days remaining. I wanted to ensure we have addressed all your concerns satisfactorily. If there are any additional points or feedback you'd like us to consider, please let us know. Your insights are invaluable to us, and we're eager to address any remaining issues to improve our work.
>
> Thank you for your time and effort in reviewing our paper.

---

### Author Response · Authors · 2025-12-02
**Overview Comment by Authors**

Dear Reviewers, AC, SAC, and PC,

We sincerely thank the ACs and reviewers for their time, effort, and constructive input throughout the review process. Below, we summarize the discussion period and highlight the importance of our work, so that ACs, reviewers, and future public readers can better understand the contributions of this research.

**Overall Reviewer Attitudes**

We are delighted that during the rebuttal period, **three reviewers (Reviewer grmy, Reviewer QVGQ, Reviewer i18D) acknowledged our rebuttal and subsequently raised their scores for our paper (finally rating scores are 6,6,6,6)**. All three reviewers expressed strong satisfaction with our rebuttal response, stating that "most of my concerns have been addressed" and decisively raised their initial rating scores. Reviewer i18D raised their score from 4 to 6 on Nov.24. Reviewer QVGQ raised their score from 4 to 6 on Nov.27. Reviewer grmy raised their score from 4 to 6 on Nov.28. All rebuttal details are provided under each reviewer's thread.

**Importance and Summary of This Research**

We are encouraged to see that the reviewers consistently recognized several key strengths of our work, including:
- Critical and practical challenge: Reviewers all think our topic is an important, critical, and practical challenge in LLM training (Reviewer grmy, Reviewer QVGQ), and is relevant and timely for large-scale LLM adaptation (Reviewer i18D).
- Sufficient theory: We provide some theoretical proofs under the premise of a well-formalized bi-level optimization description (Reviewer c1Z3, Reviewer grmy).
- Comprehensive Experimental Setup: We also conducted comprehensive experiments to evaluate IDEAL's capability across multiple diverse domains, model size and model types. (Reviewer QVGQ, Reviewer grmy)

**Addressed Key Concerns**

We have replied to all concerns/questions through clarifications, empirical corrections, and manuscript revisions (highlighted in blue font in the revised manuscript). We also clarified several key points to address the reviewers' concerns:
1. Novelty Clarification: In response to concerns about the novelty of our work, we have provided comprehensive justifications from multiple perspectives, including mathematical modeling, algorithmic design, and practical applicability, demonstrating that our method is both innovative and practically viable.
2. KFAC Efficiency & Resource Use: Regarding reviewers’ questions about KFAC approximation errors and computational costs, we derived a theoretical error bound (highlight in Appendix C) and presented empirical results to validate the efficiency and cost-effectiveness of our approach (In Reviewer c1Z3 weakness 2, Reviewer grmy weakness 1 and Reviewer QVGQ weakness 3).
3. Comparative Analysis & Paper Structure: We further enhanced the paper by refining comparisons with baseline methods and adjusting the overall structure to improve clarity and completeness (highlight in Sec.4.1).

Once again, we sincerely thank all reviewers for their recognition and valuable suggestions. We believe that our responses address the raised concerns effectively, and we hope the program committee (reviewers, ACs, SACs, PCs) will take them into account in the next phases.

Sincerely,

Authors

---

### Meta-Review · Area_Chair_mX4G · 2025-12-17

**Summary:**

This paper investigates the underexplored role of data composition in mixture supervised fine-tuning for large language models and proposes IDEAL, a gradient-based framework for adaptively balancing domain-wise data volumes. Rather than focusing on data quality, IDEAL dynamically adjusts the training mixture according to each domain’s contribution to downstream performance, leading to better alignment across multiple capabilities. Experiments show consistent and meaningful gains over uniform allocation, with clear improvement in multi-task evaluation.

It received comments from four reviewers. The main concerns are reflected in (1) the computational costs of the proposed method are huge; (2) the technical contribution of the work is weak; (3) experimental results are unclear; (4) the verification scenarios are not rich enough. After the rebuttal, the reviewers reached a consensus and considered that the paper could meet the acceptance line. Therefore, the AC recommends acceptance.

**Reviewer Concerns:**

The concerns of the four peer reviews were similar in many ways. Reviewers raised questions about the experiments, method cost issues, and unclear descriptions. A detailed rebuttal was provided subsequently. Three reviewers raised the score to be positive. The AC checks the paper, questions, and answers, and thinks that the main concerns have been addressed by the rebuttal.

**Reviewer Scores:**

- **Reviewer c1Z3.** The main concerns include (1) discussions about previous works; (2) the computational cost of the method. The rebuttal includes detailed explanations. Therefore, if the reviewer had been able to participate fully in the discussion, the score would be positive.

- **Reviewer grmy.** The main concerns include (1) the computational cost of the method; (2) the issue of Hessian approximation; (3) confusing reported results. The rebuttal includes detailed explanations and results to the concerns. Therefore, if the reviewer had been able to participate fully in the discussion, the score would be positive. This actually was acknowledged by the reviewer.

- **Reviewer QVGQ.** The concerns include (1) the limited technical novelty; (2) the computational cost of the method; (3) the theory limitation. Both results and explanations are provided. Therefore, the reviewer would be positive about the submission. This actually was acknowledged by the reviewer.

- **Reviewer  i18D.** The reviewer was concerned about (1) the computational cost of the method; (2) the issue of Hessian approximation; (3) not comprehensive evaluations. The AC checked the feedback provided by the authors. It is overall convincing. Therefore, the reviewer would be positive about the submission. This was also acknowledged by the reviewer.

---

### Decision · Program_Chairs · 2026-01-26

Accept (Poster)